## RESEARCH ARTICLE

# Functional dissection of H3K4 methyltransferases reveals distinct catalytic and non-catalytic roles in *C. elegans* development

Benedetta Attianese[1,*], Hua Wang[2,3,4], Katrine Madsen[1,§], Mandoh Zeijdner[1,¶], Kristian Helin[1,4,5], Steffen Abay-Nørgaard[1,‡,**] and Anna Elisabetta Salcini[1]

## ABSTRACT

The KMT2 class of histone methyltransferases regulates methylation of histone 3 lysine 4 (H3K4), a conserved post-translational modification associated with active transcription. However, previous studies have highlighted catalytic-independent functions of KMT2 members and questioned the influence of H3K4 methylation on gene expression. Here, we address this by generating catalytically inactive mutants of SET-2 and SET-16, the two KMT2 members in *Caenorhabditis elegans*. Through chromatin analysis, we determined the effect of SET-2 and SET-16 catalytic activities on H3K4me3 deposition and identified shared and distinct targets. Gene expression profiling showed that simultaneous inactivation of SET-2 and SET-16 catalytic activities results in gene deregulation independent of H3K4me3 status at transcription start sites. Finally, we examined the relevance of SET-2 and SET-16 catalytic activity on phenotypes identified in null mutants and found that SET-2 catalytic activity is essential for proper somatic development, whereas SET-16 enzymatic activity has cell type-specific roles. Interestingly, animals lacking SET-2 and SET-16 catalytic activity are viable and fertile under normal growth conditions. Our results reveal catalytic-dependent and -independent roles of KMT2 members, and that combined loss of SET-16 and SET-2 is compatible with life in *C. elegans*.

KEY WORDS: KMT2, H3K4 methylation, Transcription, *C. elegans*

## INTRODUCTION

Methylation of histone 3 lysine 4 (H3K4) is one of the most studied evolutionarily conserved post-translational modifications because of its implication in transcriptional regulation and disease

[1]Biotech Research and Innovation Centre (BRIC), University of Copenhagen, Ole Maaløes Vej 5, 2200 Copenhagen N, Denmark. [2]Peking University International Cancer Institute, Peking University Cancer Hospital and Institute, State Key Laboratory of Molecular Oncology, Peking University Health Science Center, Beijing 100191, China. [3]Department of Biochemistry and Molecular Biology, School of Basic Medical Sciences, Peking University Health Science Center, Beijing 100191, China. [4]Cell Biology Program and Center for Epigenetics Research, Memorial Sloan Kettering Cancer Center, New York, NY 10065, USA. [5]Division of Cell and Molecular Biology, The Institute of Cancer Research, London SW7 3RP, UK.
*Present address: Department of Molecular Biology and Genetics, Aarhus University, Denmark. ‡Present address: Department of Biotechnology and Biomedicine, Technical University of Denmark (DTU), Kongens Lyngby, Denmark. §Present address: Novo Nordisk Center for Biosustainability, Technical University of Denmark (DTU), Kongens Lyngby, Denmark. ¶Present address: Donders Center for Neuroscience, Radboud University, Nijmegen, The Netherlands.

**Author for correspondence (snoerg84@gmail.com)

B.A., 0000-0002-7760-1731; M.Z., 0000-0002-7427-9345; K.H., 0000-0003-1975-6097; S.A.-N., 0000-0002-0094-8112; A.E.S., 0000-0001-5828-2512

(Faundes et al., 2018; Rao and Dou, 2015; Shilatifard, 2012; Vallianatos and Iwase, 2015; Wang and Helin, 2024). H3K4 can be mono- (H3K4me1), di- (H3K4me2) and tri- (H3K4me3) methylated, with H3K4me3 enriched at transcription start sites (TSSs) of actively transcribed genes, and H3K4me1 predominantly found at enhancer regions (Barski et al., 2007; Bernstein et al., 2005; Cano-Rodriguez et al., 2016; Guenther et al., 2007; Heintzman et al., 2007; Lauberth et al., 2013; Santos-Rosa et al., 2002; Soares et al., 2017; Vermeulen et al., 2007; Zentner et al., 2011). Although the enzymes catalysing H3K4 methylation are conserved during evolution, their number increases from unicellular to multicellular organisms. *Saccharomyces cerevisiae* contains a single methyltransferase, Set1, which deposits all methylated forms of H3K4 (Dehé et al., 2006; Miller et al., 2001; Pokholok et al., 2005; Roguev, 2001), *Drosophila melanogaster* has three homologues of Set1 (Set1, Trx and Trr) (Mohan et al., 2011), and mammals have three pairs of methyltransferases: SETD1A/B, related to dSet1 and responsible for the majority of H3K4me3 deposition (Lee et al., 2007; Sze et al., 2020; Wu et al., 2008), MLL1/2, related to Trx, depositing H3K4me2/3 at developmental genes (Denissov et al., 2014; Hu et al., 2013; Rickels et al., 2016), and MLL3/4, related to Trr, depositing the majority of H3K4me1 at enhancer elements (Heintzman et al., 2009, 2007; Herz et al., 2012; Hu et al., 2013). These methyltransferases, also known as the KMT2 family, individually act in the context of multi-subunit SET1/COMPASS complexes interacting with proteins essential for supporting the methyltransferase (Shilatifard, 2012).

H3K4me3 is regarded as an activating post-translational modification (Bernstein et al., 2002; Pokholok et al., 2005; Santos-Rosa et al., 2002; Schneider et al., 2004; Strahl et al., 1999), and it is believed to promote transcription by recruiting PHD-containing proteins involved in transcription initiation (Lauberth et al., 2013; Li et al., 2006; Vermeulen et al., 2010, 2007) and by regulating RNApol II pausing and RNA elongation (Wang et al., 2023). Yet some studies have challenged the instructive role of H3K4 methylation in transcription (Howe et al., 2017; Morgan and Shilatifard, 2023). Indeed, ablation of Set1 in yeast results in loss of all H3K4 methylation with a minimal impact on gene expression or in enhanced transcription (Guillemette et al., 2011; Margaritis et al., 2012; Venkatasubrahmanyam et al., 2007). Furthermore, the replacement of H3K4 with non-methylatable residues in all canonical and variant H3 proteins has a minor effect on transcription in *Drosophila* (Hödl and Basler, 2012). Recent studies have also revealed that some of the KMT2 members have catalytic-independent biological functions (Morgan and Shilatifard, 2023; Van et al., 2024). For example, loss of SETD1A, but not of its catalytic activity, inhibits mouse embryonic stem cell proliferation (Bledau et al., 2014; Fang et al., 2016; Sze et al., 2017). Likewise, SETD1A promotes leukaemic cell survival by regulating the DNA damage response, independently of its enzymatic activity

(Hoshii et al., 2018). Embryonic stem cells with catalytically inactive MLL3/MLL4 differentiate normally (Cao et al., 2018; Xie et al., 2023) and show minor transcriptional changes, in comparison to null mutants (Dorighi et al., 2017). In agreement with these findings, in *Drosophila* the presence, but not the enzymatic activity, of Trr is required for viability (Rickels et al., 2017). Thus, the relevance of the catalytic activity of the KMT2 members on biological functions remains an open question.

With only two KMT2 members, SET-2 and SET-16, the nematode *C. elegans* is an amenable, genetically tractable model to investigate the biological impact of their catalytic activities at an organism level. SET-2 and SET-16 show homology with SETD1-like and MLL-like proteins, respectively, and act in distinct COMPASS complexes (Beurton et al., 2019). Previous studies have indicated that SET-2 is responsible for H3K4me2/3 deposition, transcriptional regulation and normal development (Abay-Nørgaard et al., 2020; Beurton et al., 2019; Greer et al., 2010; Han et al., 2017; Herbette et al., 2017; Robert et al., 2014), and its catalytic activity is required for fertility (Caron et al., 2022). However, the impact of SET-2 catalytic activity on the genome-wide distribution of H3K4me3 and transcription has not been systematically investigated. Much less is known about SET-16, as *set-16* loss is incompatible with life (Li and Kelly, 2011; Vandamme et al., 2012) and the impact of SET-16 on H3K4 methylation, transcription and development (Ding et al., 2018; Fisher et al., 2010; Vandamme et al., 2012) has been established mainly by reduction using RNA interference (RNAi) approaches.

Here, we determine the expression pattern of SET-2 and SET-16 and their contribution to H3K4me3 deposition and to gene expression. In addition, we identify catalytic-dependent and -independent biological functions of SET-2 and SET-16 and show that loss of the majority of H3K4me3, achieved by abolishing SET-2 and SET-16 activities simultaneously, is compatible with development and reproduction in *C. elegans* under optimal environmental conditions.

## RESULTS
### Catalytic activities of SET-2 and SET-16
To investigate the enzymatic activity of SET-2 and SET-16, we generated catalytically inactive mutants of SET-2 and SET-16, *set-2(zr1504)* and *set-16(zr1804)* (Fig. 1A), in which a conserved histidine in the SET-domain signature motif (RFINHXCXPN), required for catalytic activity (Dillon et al., 2005; Kwon, 2003; Rizzardi et al., 2012; Sharaf et al., 2022; Trievel et al., 2002; Wilson et al., 2002), was replaced with a lysine (H1147K and H2410K, respectively) (Fig. 1A,B). This amino acid change in SET-2 has been shown to affect H3K4me2/3 levels without compromising the formation of the SET-2 COMPASS complex (Abay-Nørgaard et al., 2020; Caron et al., 2022). As expected, the substitution in SET-2 led to a strong reduction of H3K4me3 levels, a decrease of H3K4me2 and only minor changes in H3K4me1 levels. The effects were comparable to those detected for *set-2(tm1630)* and another null mutant allele previously described, *set-2(bn129)* (Abay-Nørgaard et al., 2020; Li and Kelly, 2011; Xiao et al., 2011), confirming that the amino acid substitution introduced in the catalytic domain of *set-2(zr1504)* effectively compromises the catalytic activity of SET-2 (Fig. 1C,D, Fig. S1A).

In *set-16(zr1804)* animals, carrying an analogous mutation, we observed a reduction of H3K4me1/2 and minimal, if any, effect on H3K4me3 levels (Fig. 1C,D, Fig. S1A). The generation of *set-16(zr1984)* animals, carrying another SET domain mutation (R2389W), confirmed this result (Fig. S1B,C). The decrease of

H3K4me1 in *set-16(zr1804)* animals was also confirmed by chromatin immunoprecipitation (ChIP) (Fig. S3). The unexpected viability of *set-16(zr1804)* animals allowed the generation of a double mutant in which the catalytic activities of both KMT2 members were compromised. *set-2(zr1504) set-16(zr1804)* double-mutant animals showed a strong depletion of H3K4me2/3 and a reduction of H3K4me1 levels (Fig. 1C,D, Fig. S1).

To investigate the effect of loss of both SET-2 and SET-16 catalytic activities in different tissues, we analysed H3K4 methylation in *set-2(zr1504) set-16(zr1804)* double-mutant animals by immunofluorescence. In double-mutant adult animals, the H3K4me3 signal was below detection in germ cells of the mitotic and pachytene regions (Fig. 2A,B), and in intestinal cells and somatic sheath cells (Fig. 2E,F), but still evident in the diakinesis region of the germline (Fig. 2B). H3K4me2 appeared to be lost in the germline regions (Fig. 2C) but present in intestinal cells and somatic sheath cells (Fig. 2E,F). Finally, the level of H3K4me1 was less affected in the double mutant, both in germ cell compartments and somatic cells (Fig. 2D-F). Similarly, in *set-2(zr1504) set-16(zr1804)* double-mutant embryos, H3K4me2/3 was below detection levels whereas H3K4me1 was still detectable, although at a lower intensity compared to wild type (Fig. 2G). In this context, it should be noted that the H3K4me1 antibody used in these assays has been validated (http://compbio.med.harvard.edu/antibodies/) and previously utilised for western blot and immunofluorescence (e.g. Greer et al., 2010).

Overall, these results show that SET-2 and SET-16 both contribute to the three different forms of methylation, with SET-2 mainly responsible for H3K4me2/3 deposition and SET-16 primarily for H3K4me1/2. In addition, the substantial effect of the loss of catalytic activity of SET-2 and SET-16 on H3K4 methylation suggests a broad expression of these proteins.

### Expression patterns of SET-2 and SET-16
To analyse the spatial-temporal expression of SET-2 and SET-16 and to correlate it with their effect on H3K4 methylation, we introduced an N-terminal GFP tag in their respective loci by CRISPR/Cas9 genome engineering. Because the constructs retained endogenous regulatory elements, the tagged proteins are expected to reflect native expression patterns (Dickinson et al., 2013; Kim et al., 2014). Our findings confirm and extend previous reports of broad SET-2 and SET-16 expression (Engert et al., 2018; Simonet et al., 2007). We analysed the expression of SET-2 and SET-16 in germ cells and somatic tissues. In adult germ cells, SET-2 was expressed in the germ cell nuclei of all regions of the germline (Fig. S2A, left). In contrast, SET-16 expression was restricted to germ cells located at the proximal part of the gonad, mainly in the diakinesis region (Fig. S2A, right). In germ cells (Z2/Z3, identified by PGL-1 staining), SET-2 and SET-16 expression was lower than in the neighbouring embryonic somatic cells, where SET-2 and SET-16 were consistently detected (Fig. S2B). We also observed a broad nuclear expression of SET-2 and SET-16 among somatic adult cells, including intestinal cells, neurons and distal tip cells (Fig. S2A,C). Notably, SET-2 and SET-16 were present during early embryogenesis, suggesting that they are maternally provided to the zygote (Fig. S2C). Overall, these results indicate that SET-2 and SET-16 are broadly expressed both in embryos and adults with overlapping expression in somatic tissues but with distinct distribution in adult germ cells.

### Contribution of SET-2 and SET-16 catalytic activities to genome-wide H3K4me3 deposition
To test the contribution of the catalytic activity of SET-2 and SET-16 to H3K4me3 deposition at the genome-wide level, we performed

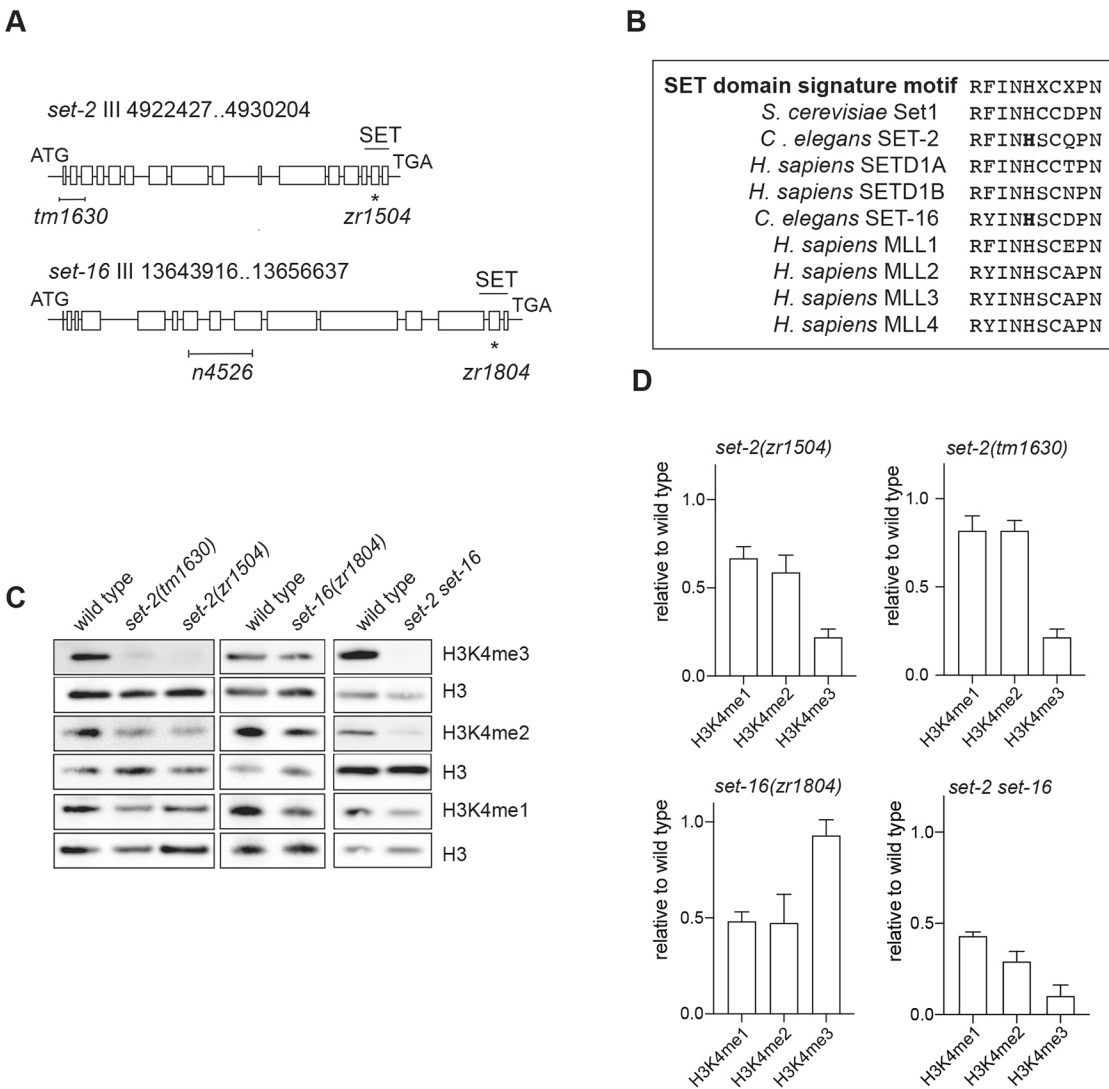

**Fig. 1. SET-2 and SET-16 catalytic activities.** (A) Genomic structure of *set-2* and *set-16* genes. H-shaped bars indicate *set-2(tm1630)* and *set-16(n4526)* deletion mutants. Asterisks indicate point mutations in a conserved region of the SET domain (H1147K and H2410K) carried by *set-2 (zr1504)* and *set-16(zr1804)* alleles, respectively. The position of the SET domain is indicated. (B) SET-domain signature motif. Amino acids mutated in *set-2 (zr1504)* and *set-16(zr1804)* alleles are indicated in bold. (C) Representative western blots of lysates from mixed embryos of indicated genotypes, probed with antibodies specific for H3K4me1/2/3. H3 is used as loading control. *set-2 set-16* indicates *set-2(zr1504) set-16(zr1804)* double-mutant animals. (D) Western blot quantification of H3K4me1/2/3 levels in the indicated genotypes. Levels are reported relative to wild-type animals and are presented as mean±s.e.m. from three independent biological experiments. *n*=3 per genotype.

H3K4me3 ChIP in wild-type, *set-2(zr1504)*, *set-16(zr1804)* and double-mutant *set-2(zr1504) set-16(zr1804)* animals at the young adult stage. Consistent with previous analyses in other organisms (Barski et al., 2007; Wang et al., 2008) and in *C. elegans* (Jänes et al., 2018; Liu et al., 2011), H3K4me3 was enriched at the 5′ end of the genes, around the TSSs (Fig. 3A, top; Fig. S4A) in wild-type animals. We also detected H3K4me3 at a subset of putative enhancers, as previously reported (Jänes et al., 2018) (Fig. 3A, bottom). As expected, the H3K4me3 signal at regulatory elements was strongly affected in the *set-2(zr1504)* mutant (Fig. 3A, Fig. S4A). In *set-16(zr1804)* mutants, we observed a slight reduction in H3K4me3 (Fig. 3A, Fig. S4A), barely detected by western blot assays performed in embryo lysates. Whether this reduction is stage-specific or reflects limitations in western blot sensitivity is unclear; however, this reduction is consistent with results showing that the mammalian SET-16 homologue MLL2 contributes to H3K4me3 deposition at promoters (Sze et al., 2020).

Importantly, the H3K4me3 signal was nearly abolished in the *set-2(zr1504) set-16(zr1804)* double mutant, at TSSs and putative enhancers (Fig. 3A), indicating that SET-2 and SET-16 jointly catalyse the majority of H3K4me3 at transcriptional regulatory regions.

To identify potential targets of SET-2 and SET-16, we searched for annotated genes showing H3K4me3 decrease at the TSS (log2FC<−1) in single and double mutants, compared to wild type (Fig. 3B, Table S1). Using this criterion, 2772 and 237 genes were identified in *set-2(zr1504)* and *set-16(zr1804)* mutants, respectively. In *set-2 set-16* double mutant, a larger number (3353) of loci showed decreased H3K4me3 levels at TSSs. Notably, 153 of the 237 SET-16 targets (65%) overlapped with SET-2 targets and more than 600 genes lost H3K4me3 at the TSS specifically in the double mutant, suggesting both overlapping and non-redundant functions of SET-2 and SET-16 enzymatic activities on H3K4me3 regulation at specific loci (Fig. 3C). K-means clustering identified four gene clusters

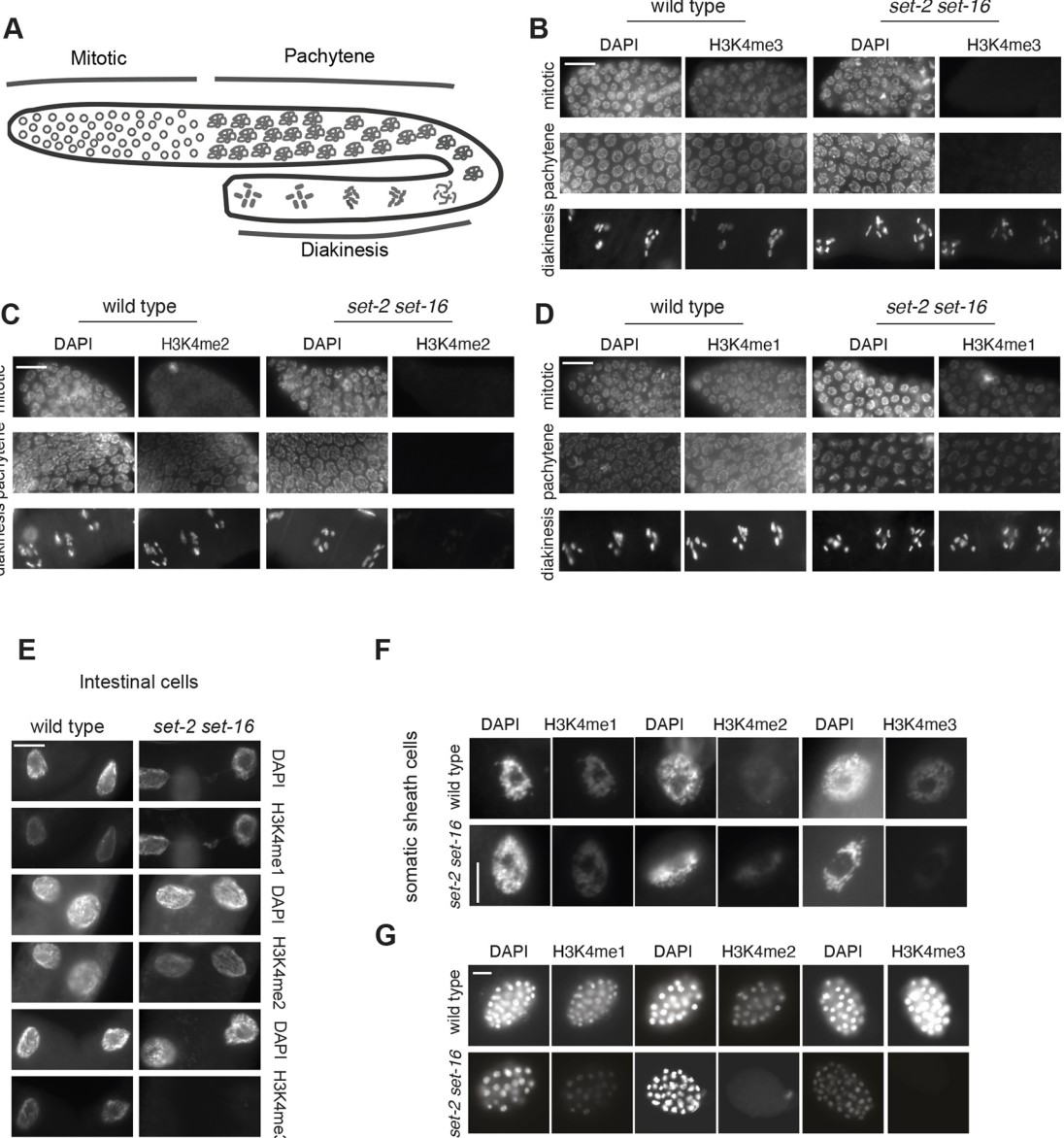

**Fig. 2. H3K4 methylation in the *set-2 set-16* double mutant.** (A) Schematic of the major compartments of the germline. (B-D) Representative images of different regions of the germline, extracted and stained with DAPI and H3K4me1/2/3, of wild type and the *set-2(zr1504) set-16(zr1804)* double mutant. (E,F) Representative images of intestinal cells (E) and somatic sheath cells (F) stained with DAPI and H3K4me1/2/3 in wild-type and *set-2(zr1504) set-16(zr1804)* double-mutant animals. (G) Representative images of wild type and *set-2(zr1504) set-16(zr1804)* double-mutant embryos, stained with DAPI and H3K4me1/ 2/3. Images are representative of five samples. Scale bars: 10 µm (B-E,G); 5 µm (F).

(Table S1) with distinct H3K4me3 patterns among single and double mutants (Fig. 3D,E). Gene ontology (GO) analysis of genes falling in different clusters is shown in Fig. S5.

Overall, these results illustrate the prominent effect of SET-2 and of SET-16 catalytic activities on H3K4me3 at gene regulatory elements and identify putative SET-2 and SET-16 targets in adult animals.

## Impact of SET-2 and SET-16 catalytic activities on gene expression

To assess the role of SET-2 catalytic activity in gene expression, we performed mRNA sequencing (RNA-seq) in both a null mutant, *set-2(tm1630)*, and animals with a catalytically inactive allele, *set-2(zr1504)*, at the young adult stage. We identified 2236 differently expressed (DE; abs log2FC >1, *P*adj<0.01) genes (1660 upregulated and 576 downregulated) in *set-2(tm1630)* (Table S2,

Fig. S6A), compared to wild type. Inactivation of SET-2 catalytic activity had a smaller but still prominent effect on gene expression, with 1008 DE genes in *set-2(zr1504)* (757 upregulated and 251 downregulated) (Fig. S6B, Table S2), the majority of which (78%) overlapped with DE genes in *set-2(tm1630)* animals (Fig. 4A). This result, while validating the robust impact of the catalytic activity in transcription regulation, also suggests that a portion of deregulated genes observed in absence of *set-2* does not directly depend on its catalytic activity. Similarly, transcriptome profiling of *set-16(zr1804)* identified 1136 DE genes (Fig. S6C, Table S2), the majority of which (991/1136) were upregulated. Interestingly, *set-2(zr1504)* and *set-16(zr1804)* mutants shared a considerable number of DE genes (489 genes) (Fig. 4B), suggesting partially overlapping functions of these methyltransferases in terms of gene expression regulation. In agreement, GO analyses of *set-2* and

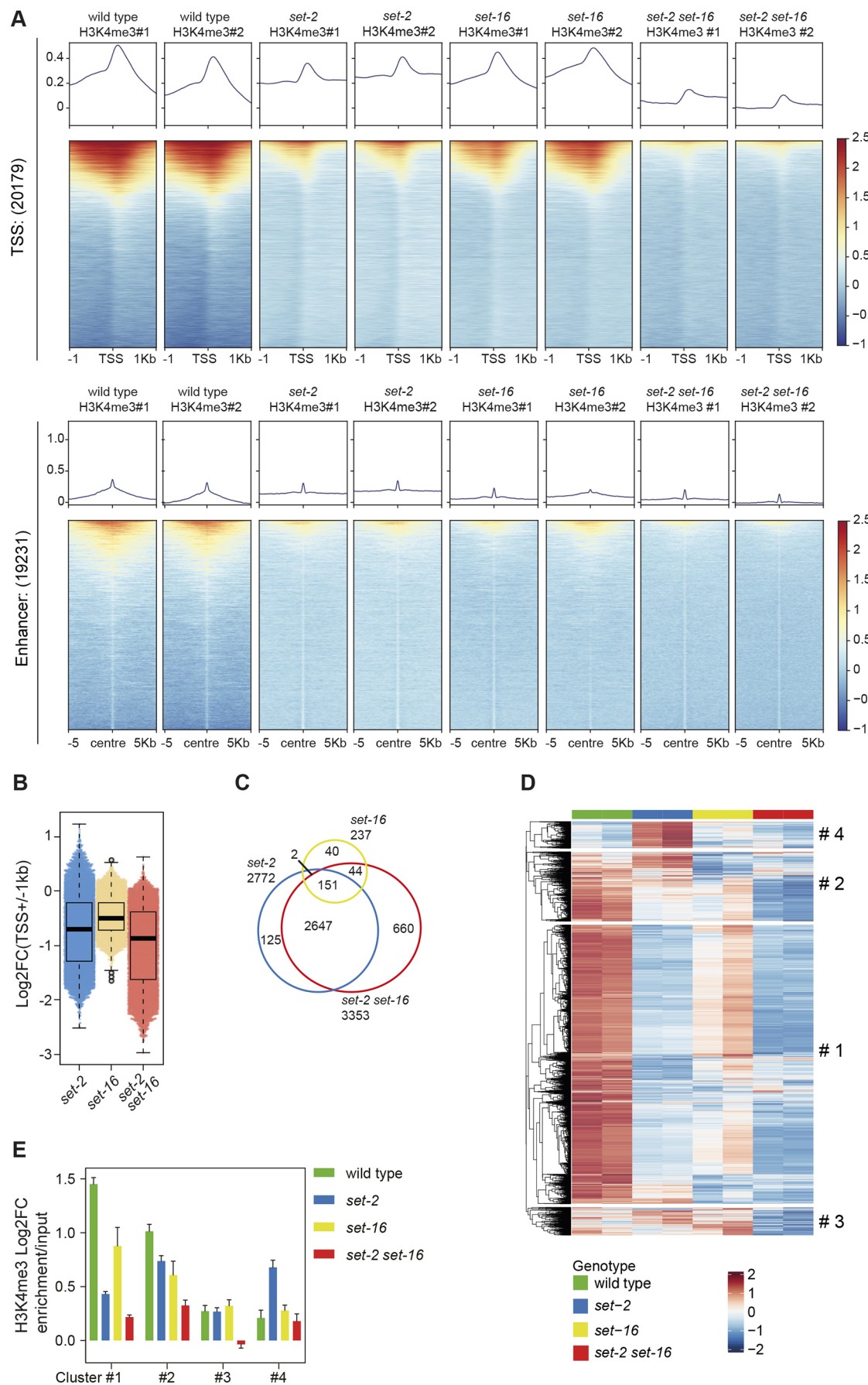

**Fig. 3.** See next page for legend.

**Fig. 3. Genome-wide analysis of SET-2 and SET-16 catalytic activities on H3K4me3.** (A) Heatmaps of H3K4me3 ChIP-seq enrichment at TSSs (±1 kb) (top) and putative enhancers taken from Jänes et al. (2018) (±5 kb from centre) (bottom) in wild type, *set-2(zr1504)* single mutant, *set-16(zr1804)* single mutant and *set-2(zr1504) set-16(zr1804)* double mutant. Two biological independent experiments are shown. *n*=4 per genotype. (B) Beeswarm plot based on log2FC of H3K4me3 enrichment at TSSs of the indicated genotypes, compared to wild type. The box plots indicate the median (centre line), the third and first quartiles (box limits) and 1.5×IQR above and below the box (whiskers). *n*=4 per genotype. (C) Venn diagram reporting the number and overlap of loci with H3K4me3 reduction at the TSS (log2FC<−1) in the indicated genotypes. (D) K-mean clustering (K=4) based on H3K4me3 level in the indicated genotypes. (E) Average bar plots reporting, for each cluster identified in D, the log2F value of the mean enrichment in each H3K4me3 ChIP-seq sample divided by its corresponding input sample. Error bars represent s.e.m. *n*=4 per genotype.

*set-16* DE genes identified common terms, with three main domains enriched, related to collagens/moulting cycle, innate immune/stress responses, and DNA packaging (Fig. S6E). Finally, we analysed the transcriptome of the *set-2(zr1504) set-16(zr1804)* double mutant. Compared to single mutants, we observed a slightly larger number of DE genes (1951, with 1541 upregulated and 410 downregulated) (Fig. S6D, Table S2) and, despite a sizable overlap with single mutants (Fig. 4C) reflected also in the GO analysis (Fig. S6E), 759 genes were aberrantly regulated specifically in the double mutant, indicating that the ablation of both H3K4 methyltransferases has a greater impact on gene expression. By clustering DE genes by K-mean, we identified five clusters of genes expression of which significantly changed in the *set-2(zr1504) set-16(zr1804)* double mutant compared to the single mutants (Fig. 4D, Table S2). Clusters 2, 3 and 4 contain genes for which expression in the double mutant was less affected compared to single mutants, suggesting an antagonistic function of *set-2* and *set-16* catalytic activities. Clusters 1 and cluster 5, by contrast, contain genes for which expression levels were further changed in the double mutant compared to the single mutants, both in term of upregulation (cluster 5) and downregulation (cluster 1), suggesting a synergistic effect of the catalytic activities of SET-2 and SET-16. Finally, clusters 1 and 2 contain genes that were downregulated in the *set-2* mutant, the *set-16* mutant and the *set-2 set-16* double mutant, compared to wild-type animals. GO analyses of genes in clusters 1 and 2 identified lipid metabolism and defence response to pathogens as categories significantly enriched (Fig. 4E), indicating that the catalytic activity of KMT2 members is required to promote the expression of genes involved in these biological processes, which also have been proposed in previous studies (Ding and Wang, 2015; Ding et al., 2018; Han et al., 2017). Altogether, our RNA-seq analyses indicate that both SET-2 and SET-16 catalytic activities contribute to gene expression and, as previously reported in embryos (Abay-Nørgaard et al., 2020), loss of H3K4me3 mainly results in gene upregulation in *C. elegans*.

To examine further the connection between H3K4me3 and transcription, we assessed the correlation between H3K4me3 levels and gene expression, by plotting for each gene the RNA-seq log2 fold change versus the H3K4me3 ChIP-seq log2 fold change (±1 kb around the TSS) in *set-2* and *set-16* single and double mutants. Strikingly, no overall correlation between H3K4me3 status and gene expression was observed in any of the genetic backgrounds tested (Fig. 4F) and most of the genes with significantly decreased levels of H3K4me3 at TSSs were not downregulated. Surprisingly, several of the genes that showed an increased expression in the double mutants were associated with decreased H3K4me3 levels. These findings suggest that even a strong reduction in H3K4me3, caused

by loss of SET-2 and SET-16 catalytic activity, does not impair transcription in *C. elegans*.

### Effects of acute depletion of SET-2 and H3K4me3 on gene expression

The previous transcriptome analyses were performed on mutant animals grown for multiple generations, allowing potential adaptation to H3K4me3 loss. To investigate the effects of an acute H3K4me3 loss, we took advantage of an auxin-inducible degradation (AID) system to rapidly deplete SET-2 and SET-16. Attempts to generate a functional SET-16-AID strain were unsuccessful due to embryonic and larval lethality, even in the absence of auxin, precluding further analysis. We therefore focused on SET-2, the major contributor to H3K4me3, and created a strain expressing N-terminally AID/GFP-tagged SET-2 in a background that enables tissue-wide, auxin-dependent degradation (Zhang et al., 2015). L4 animals treated with auxin for 6 h, until the young adult stage was reached, showed no detectable GFP::SET-2 signal in somatic and germ cells (Fig. 5A). Importantly, in this condition, H3K4me3 was strongly reduced, while H3K4me1/2 levels remained largely unchanged (Fig. 5B). Consistently, H3K4me3 ChIP sequencing (ChIP-seq) showed decreased levels of H3K4me3 at TSSs (Fig. 5C,D, Fig. S4C) and putative enhancers (Fig. 5C) after 6 h of auxin treatment, compared to control, suggesting a rapid turnover of H3K4me3. We identified ~8000 peaks with significantly decreased H3K4me3 at their TSSs (log2FC<−1) at TSSs that we called 'immediate SET-2 gene targets' (Table S3). This set of genes, despite being greater in number, significantly overlapped with SET-2 gene targets identified in the *set-2(zr1504)* mutant (Fig. 5E).

To test the effect of acute loss of SET-2 and consequent H3K4me3 reduction on gene expression, we performed RNA-seq analysis of animals treated for 6 h and identified 335 DE genes (abs log2FC>1, *P*adj<0.01) (Table S2) that we called 'immediate SET-2 transcriptional targets' (Fig. 5F). These genes were equally distributed among the up- and downregulated groups. Deregulation of some of DE genes was confirmed by qPCR analyses (Fig. S7). The small number of deregulated genes after 6 h of auxin treatment is likely related to mRNA turnover and stability. However, only a negligible percentage of the immediate SET-2 transcriptional targets was aberrantly expressed in the *set-2(zr1504)* mutant (Fig. 5G), indicating that acute and chronic loss of H3K4me3 impact the expression of different genes. GO analysis of DE genes (abs log2FC>1, *P*adj<0.01) after auxin treatment is shown in Fig. 5H.

To assess the correlation between H3K4me3 and gene expression levels, we plotted for each gene the RNA-seq log2 fold change versus the H3K4me3 ChIP-seq log2 fold change (at ±1 kb of the TSS) determined in auxin-treated animals. As for the stable SET-2 and SET-16 mutants, we did not observe a correlation between changes in H3K4me3 levels at TSSs and gene expression (Fig. 5I), and the mRNA levels of most of the genes with significant decreased H3K4me3 were not reduced. As also observed for the stable mutants, a decrease in H3K4me3 levels was frequently associated with an increase in expression. These results further demonstrate the lack of correlation between H3K4me3 levels and gene expression, and may suggest that H3K4me3 is not strictly required for transcription in *C. elegans*.

### Catalytic-dependent and -independent functions of SET-2 and SET-16

To investigate the biological roles of SET-2 and SET-16 enzymatic activity in *C. elegans*, we tested whether phenotypes previously reported in null mutants were recapitulated in catalytically inactive

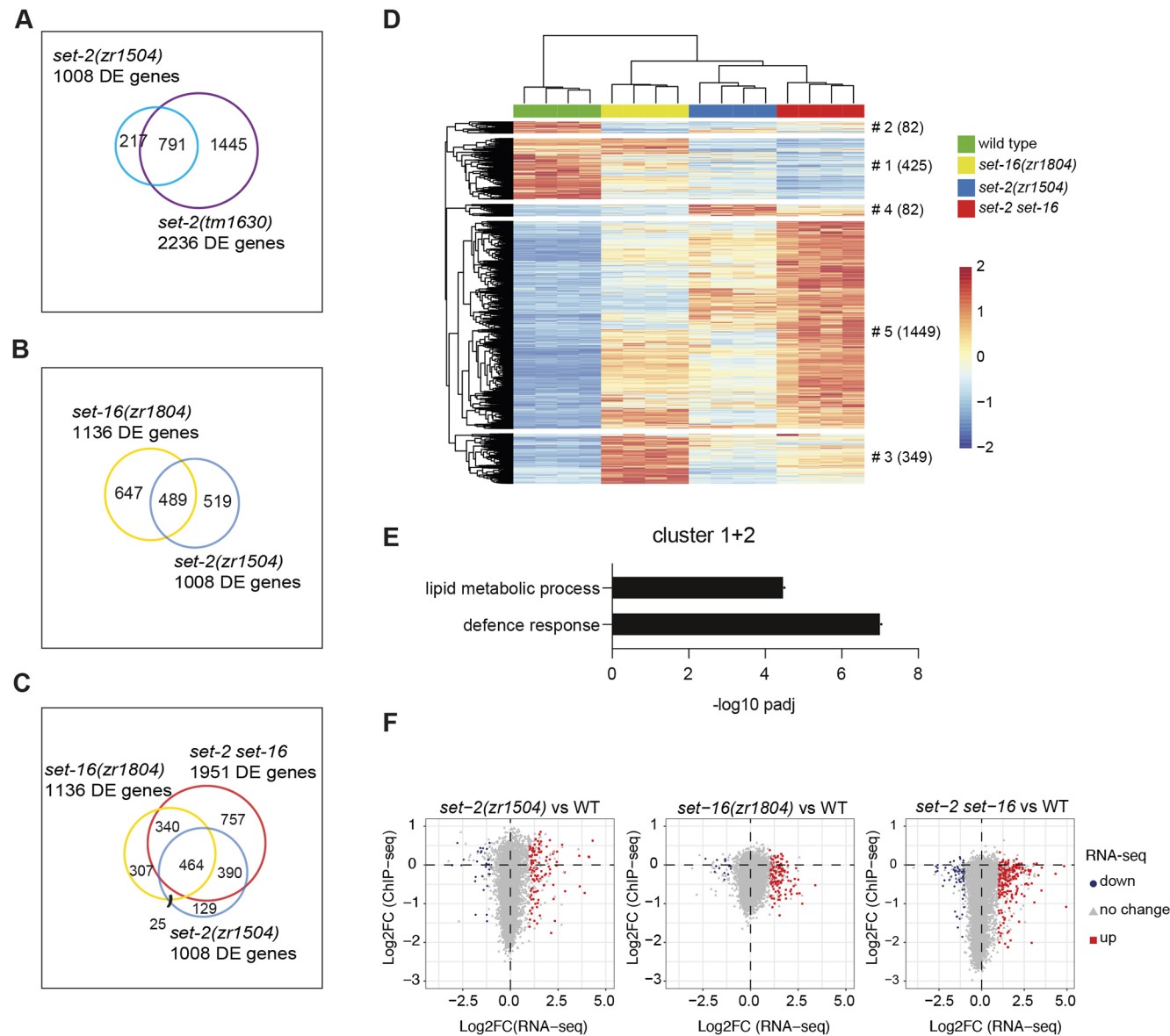

**Fig. 4. Impact of SET-2 and SET-16 catalytic activity on transcription regulation and correlation with H3K4me3 level.** (A-C) Venn diagrams showing the number and the overlap of DE genes (abs log2FC>1, $P$adj<0.01) identified in the indicated strains. (D) K-mean clustering (K=5) based on gene expression in *set-2(zr1504)*, *set-16(zr1804)* and the double mutant *set-2(zr1504) set-16(zr1804)*. Four replicas are shown. (E) GO analysis of genes in clusters 1 and 2 performed by g-profiler ($P$<0.05 with Bonferroni correction). GO terms are presented as −log10 $P$adj. (F) Correlation plots between RNA-seq and H3K4me3 ChIP-seq at TSS (±1 kb) for *set-2(zr1504)*, *set-16(zr1804)* and the double mutant *set-2(zr1504) set-16(zr1804)*.

alleles. Consistent with its broad expression, SET-2 is known to function in both somatic and germ cells. The *set-2(tm1630)* null mutant exhibits reduced fertility at 20°C and 25°C, and a progressive loss of fertility over generations at 25°C, known as the mortal germline phenotype (Li and Kelly, 2011; Robert et al., 2014) (Fig. 6A-C). Loss of SET-2 also leads to defects in PVQ axon guidance – bilateral interneurons that extend anterior axons during embryogenesis (Abay-Nørgaard et al., 2023, 2020) (Fig. 6D). These phenotypes were fully reproduced in *set-2(zr1504)* catalytically inactive mutants (Fig. 6A-D), indicating that SET-2 enzymatic activity is required for fertility and proper neuronal development. Further evaluation of other reported *set-2* phenotypes (Caron et al., 2022; Greer et al., 2011, 2010; Herbette et al., 2017) will be necessary to comprehensively define the role of its catalytic function.

SET-16 is expressed almost exclusively in somatic tissues, in line with phenotypes observed in *set-16(n4526)* null animals, including complete embryonic or larval lethality and posterior morphological abnormalities (Li and Kelly, 2011; Vandamme et al., 2012). RNAi-mediated knockdown of SET-16 also causes gonad migration defects, although this phenotype is only visible in RNAi animals that survive early lethality, possibly due to incomplete knockdown (Vandamme et al., 2012). Strikingly, animals carrying the *set-16(zr1804)* catalytically inactive allele were viable and fertile at all temperatures tested (Fig. 6A-C), and they did not show defects in gonadal migration (Fig. 6E) or in the posterior part of the body (Fig. 6F). *set-16(n4526)* null larvae also show, like *set-2* mutants, PVQ axon guidance defects (Abay-Nørgaard et al., 2020), a phenotype that was reproduced with a similar penetrance in *set-16(zr1804)* (Fig. 6D). Thus, although dispensable for viability, morphology and gonadal development, the catalytic activity of SET-16 is relevant required for proper axon guidance. These results suggest that SET-16 catalytic activity provides cell-specific functions.

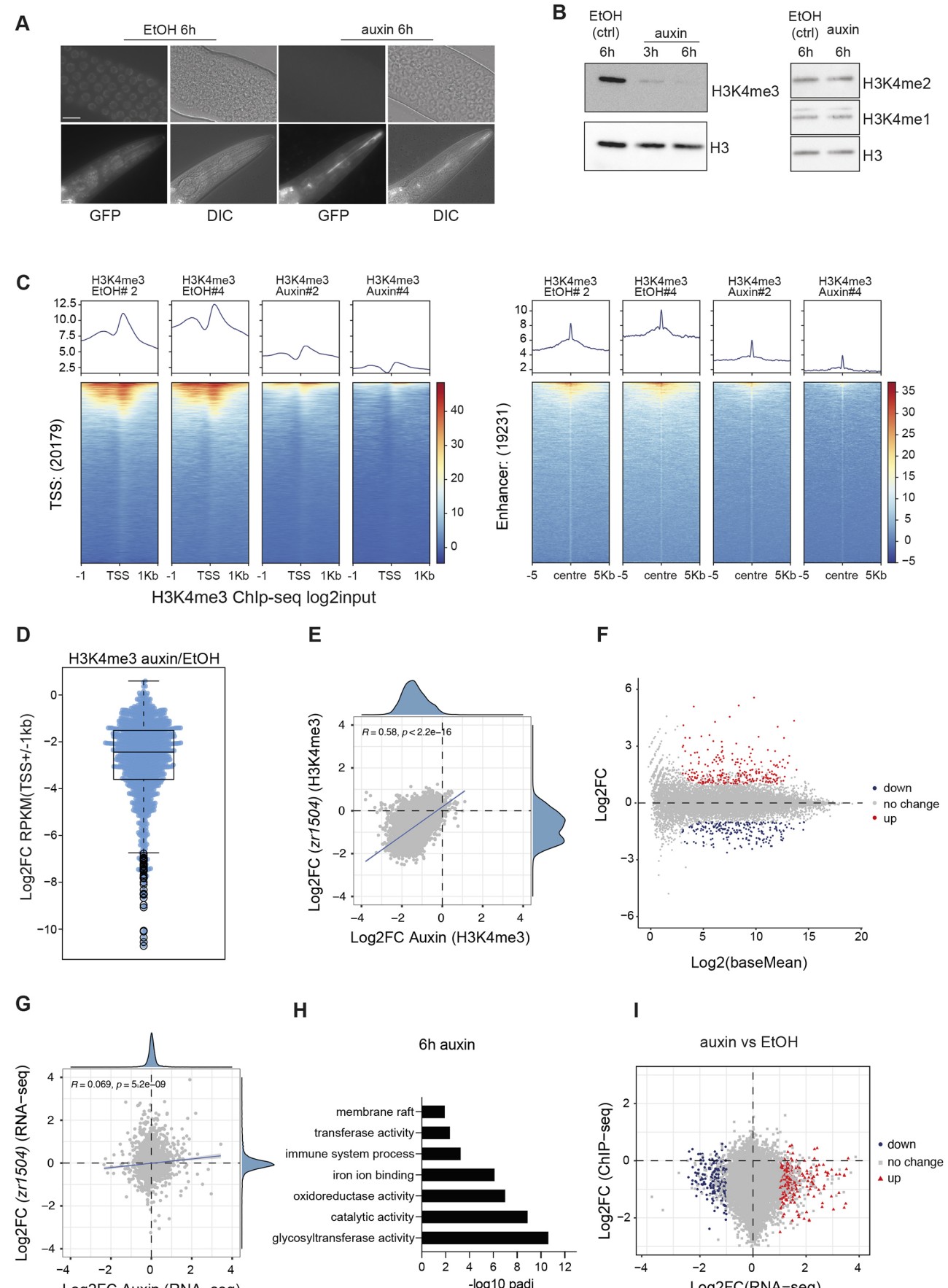

**Fig. 5.** See next page for legend.

**Fig. 5. Impact of acute depletion of SET-2 protein on gene expression and correlation with H3K4me3 level.** (A) SET-2::GFP expression in auxin-treated (1 mM; 6 h) and control (0.25% ethanol) AID/GFP-SET-2 animals, carrying a degron/GFP cassette at the 5′ end of the *set-2* gene. Representative direct fluorescence and differential interference contrast images of pachytene regions of the germline (top) and anterior part of the animal (bottom) are shown. Scale bar: 10 µm. (B) Left: Representative western blot of lysates extracted from AID/GFP-SET-2 young adult animals: control (ethanol, EtOH) or exposed for 3 and 6 h to auxin treatment, probed with H3K4me3 antibody. H3 is used as loading control. Right: Representative western blot of lysate extracted from AID/GFP-SET-2 young adult animals treated (6 h) and non-treated (EtOH) with auxin, probed with H3K4me1/2 antibodies. H3 is used as loading control. Images and blots in A and B are representative of 20 animals. (C) Heatmaps of H3K4me3 ChIP-seq signal at TSSs (±1 kb) (left) and putative enhancers (±5 kb from centre) (right) in auxin-treated (6 h) and non-treated (EtOH) animals. Two biological independent experiments are shown. *n*=4 per genotype. (D) Beeswarm plot based on the log2RPKM of H3K4me3 enrichment at TSSs in 6-h auxin-treated animals compared to control (EtOH). (E) Correlation plots between H3K4me3 ChIP-seq at TSSs (±1 kb) data from *set-2(zr1504)* mutant and auxin-treated animals (6 h). (F) MA plot showing differentially expressed genes after auxin treatment (6 h) compared to control (EtOH). (G) Correlation plot between RNA-seq data from *set-2(zr1504)* mutant and auxin-treated animals (6 h). (H) GO analysis of DE genes after 1 mM auxin treatment (6 h), compared to control (0.25% EtOH). GO analysis was performed with g-profiler (*P*<0.05 with Bonferroni correction) and GO terms are presented as −log10 *P*adj. (I) Correlation plot between RNA-seq and H3K4me3 ChIP-seq at TSS (±1 kb) data from animals treated with auxin (6 h). *n*=2 per genotype.

Importantly, the generation of *set-2(zr1504) set-16(zr1804)* double-mutant animals allowed us to test the biological effects of losing the majority of H3K4me2/3 and potential compensatory/redundant functions of SET-2 and SET-16. The double-mutant animals were viable and fertile at 20°C, with only a moderate reduction of the brood size, as observed for the *set-2(zr1504)* allele (Fig. 6A). The double-mutant animals also showed reduced brood size at high temperature (Fig. 6B) and a temperature-sensitive mortal germline (Fig. 6C), fully phenocopying the *set-2(zr1504)* mutant. Finally, loss of the catalytic activities of both SET-2 and SET-16 did not cause synergistic effects on PVQ (Fig. 6D), suggesting that *set-2* and *set-16* act in the same pathways that regulate axon guidance. In summary, our data show that the catalytic activities of SET-2 and SET-16 are not redundant. Despite a severe loss of H3K4me2/3, double mutants remain viable and fertile under standard conditions, indicating that extensive depletion of H3K4 methylation is compatible with life and reproduction in *C. elegans*, and that H3K4me3 is not strictly required for development.

## DISCUSSION
In this study, we investigated the biological relevance of the catalytic activities of SET-2 and SET-16, the only *C. elegans* members of the KMT2 family, and their contribution to H3K4 methylation deposition and gene expression.

By generating point mutations in the catalytic domain of SET-2 and SET-16, we found that SET-2 and SET-16 have overall complementary functions, with SET-2 mainly devoted to H3K4me2/3 deposition and SET-16 more actively placing H3K4me1/2. Nevertheless, SET-16 partially contributes, directly (e.g. by depositing H3K4me3) or indirectly (e.g. by controlling the amount of H3K4me1/2 available for further methylation), to H3K4me3 levels. The viability of *set-2 set-16* double-mutant animals provided us with a unique opportunity to analyse the impact of KMT2 catalytic activities on H3K4 methylation levels. We observed that H3K4me1 is largely preserved in the double-mutant

animals, both in somatic and germ cells. The residual presence of H3K4me1 in *set-2 set-16* double-mutant animals may be related to the specificity of the antibody used, but it could also suggest the presence of other methyltransferases depositing methyl groups at H3K4. Further analyses are required for their identification; SET-17 and SET-30, homologues of PRDM7/9 and KMT3C, respectively, could be potential candidates (Greer et al., 2014). Interestingly, we also revealed a dramatic decrease of H3K4me3 signal in *set-2 set-16* double-mutant animals, with one notable exception of residual H3K4me3 staining in oocytes. Similar H3K27me2/3 oocyte-persisting staining has been detected in *mes-2* mutant animals, lacking the only known H3K27 methyltransferase in *C. elegans* (Bender et al., 2004), suggesting that the chromatin methylation landscape in oocytes might depend on specific, yet-undiscovered, methyltransferases. The requirement of multiple methyltransferases for establishing the pattern of methylation in different germ cell compartments, recently analysed in detail, could be expected considering the relevance of epigenetics in gametes in maintaining germ cell immortality, in supporting totipotency in the zygote and in the initiation of embryogenesis in the absence of active transcription (Mazzetto et al., 2024).

To study the impact of SET-2 and SET-16 enzymatic activities, we investigated the occurrence and penetrance of phenotypes identified in strains carrying deletions of *set-2* and *set-16* in catalytically inactive mutant animals. The results of these analyses showed that the catalytic activity of SET-2 is essential for all tested biological functions in germ and somatic cells, whereas most of the developmental defects (posterior morphological abnormalities and aberrant gonadal migration) identified in the absence or after reduction of SET-16 are independent of its catalytic activity. Our observation that *set-16(zr1804)* animals, which carry a catalytically inactive SET-16 protein, are viable and fertile contrasts with prior findings showing that *set-16* RNAi or deletion alleles result in embryonic lethality, sterility and Dumpy phenotypes (Andersen and Horvitz, 2007; Li and Kelly, 2011; Vandamme et al., 2012). This discrepancy strongly suggests that SET-16 has essential biological functions that are independent of its methyltransferase activity. One possibility is that the SET-16 protein acts as a scaffold to recruit other COMPASS subunits or regulatory factors to chromatin. We propose that the developmental defects observed after ablation or reduction of *set-16* are related to aberrant assembly or targeting of the SET-16/COMPASS complex to chromatin. Indeed, these defects are also observed when specific (UTX-1, PIS-1) or core (WDR-5, RBBP-5) components of the SET-16 COMPASS complex are ablated or reduced (Vandamme et al., 2012). The viability of *set-16* catalytically inactive mutants allowed us to analyse the phenotypes in animals lacking the catalytic activity of the KMT2 class of methyltransferases. Surprisingly, despite being severely depleted of H3K4me2/3, double-mutant embryos are viable and able to reach adulthood. In addition, double-mutant animals are fertile, despite their germlines being depleted of H3K4me2 and H3K4me3 in mitotic and pachytene cells. These results point to subtle or compensable functional roles of H3K4me2/3 in nematode life, at least under optimal growth conditions.

As part of this study, we examined how H3K4me3 influences gene expression. Consistent with prior work, our findings show that loss of H3K4me3 does not strongly impair transcription. In *S. cerevisiae*, deletion of *Set1* eliminates H3K4 methylation but has only mild or paradoxical effects on gene expression (Guillemette et al., 2011; Margaritis et al., 2012). In *C. elegans*, transcriptome analyses across multiple mutant backgrounds have revealed that most differentially expressed genes were upregulated, further

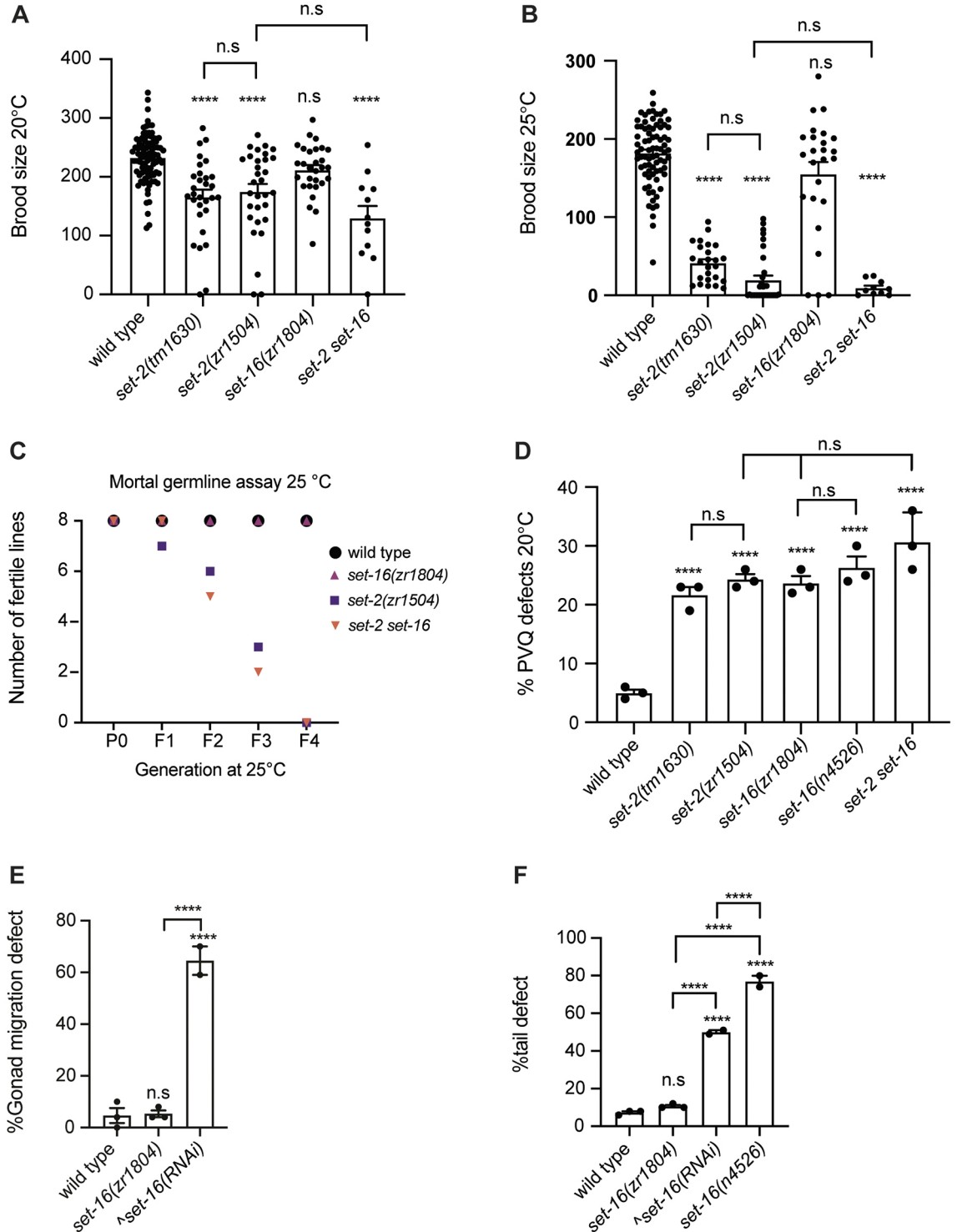

**Fig. 6. SET-2 and SET-16 catalytic activities and biological functions.** (A,B) Brood size of the indicated strains recorded at 20°C and 25°C, respectively. *set-2 set-16* indicates *set-2(zr1504) set-16(zr1804)* double-mutant animals. Each dot represents the progeny of a single animal (*n*=10). (C) Mortal germline assay of the indicated strains at 25°C. Dots designate the number of fertile lines at the indicated generation. *n*=8. (D) Percentage of PVQ axon guidance defects in the indicated strains at 20°C. Each dot represents an independent biological score of 50 adult animals. *n*=150. (E) Percentage of gonad migration defects in wild type, *set-16(zr1804)* and in wild-type animals exposed to *set-16(RNAi)*. Each dot represents an independent biological score of 40 adult animals. *n*=120. (F) Percentage of tail defects in wild type, *set-16(zr1804)*, *set-16(n4526)* and in wild-type animals exposed to *set-16(RNAi)*. Each dot represents an independent biological score of 50 adult animals. *n*=150. In A,B,D-F, data are presented as mean±s.e.m., ****P<0.0001 (one-way ANOVA with Tukey's multiple comparison test). n.s., not significant. ^Data from Vandamme et al. (2012).

supporting the suggestion that extensive H3K4me3 loss is not broadly repressive (Abay-Nørgaard et al., 2020; Beurton et al., 2019; Robert et al., 2014). Moreover, by comparing the level of H3K4me3 at TSSs and mRNA levels in *set-2* and *set-16* single- and double-mutant animals, we observed no correlation between gene expression level and H3K4me3 status. Analysis of the immediate

effects of H3K4me3 reduction on transcription, by depleting SET-2 with an auxin system, showed similar results, with most of the genes acutely depleted of H3K4me3 being not differentially expressed. Although this analysis suggest that most genes are independent of H3K4me3 for their expression, this interpretation should be taken with some caution because we have measured total mRNA levels and not *de novo* synthesised mRNAs. Therefore, cell- or tissue-specific analyses of newly synthesised mRNAs after auxin treatment are required to further dissect the relationship between H3K4me3 status and transcription at organismal level, as we recently performed in mouse embryonic stem cells (Wang et al., 2023). Intriguingly, our study shows that, despite a good correlation between the H3K4me3 profiles in SET-2 auxin-treated and *set-2* catalytically inactive mutant animals, the effect of acute depletion of SET-2 on H3K4me3 landscape is greater and the gene expression patterns in these two conditions are distinct. This result suggests that the acute loss of H3K4me3 at specific targets can be partially compensated for over time, as previously suggested (Abay-Nørgaard et al., 2023), and that most of the DE genes identified in mutant animals are likely secondary targets of SET-2, as result of adaptation programmes. It is possible that the altered transcription hinges on changes of chromatin conformation occurring after loss of SET-2 and H3K4me3. In support of this possibility, stable inactivation of COMPASS components has been shown, over generations, to lead to global alteration of H3K9 methylation (Lee et al., 2019; Robert et al., 2014). Furthermore, in *C. elegans* and mice, maternal reprogramming and transcriptional resetting can occur through parallel pathways involving histone demethylases such as LSD1/SPR-5 and the CoREST/SPR-1 complex (Carpenter et al., 2023; Spracklin et al., 2023). These systems may buffer transcription by modulating other histone marks and chromatin accessibility. Thus, further investigation aiming to identify the global changes of chromatin structure occurring in *set-2* and *set-16* mutants are required to fully understand the connection between H3K4 methylation levels and gene expression. Overall, our findings show that SET-2 and SET-16 carry out both catalytic and non-catalytic roles and that the loss of H3K4me3, even when nearly complete, is compatible with life, fertility and transcription in *C. elegans*. These results emphasise the robustness of gene regulatory networks and raise important questions about the true functional output of histone methylation marks.

## MATERIALS AND METHODS
### Animal model
*C. elegans* strains were grown at 20°C on nematode growth medium (NGM) plates seeded with *Escherichia coli* OP50 bacteria, following standard procedures (Brenner, 1974), unless otherwise stated. Double mutants were generated by standard genetic crosses. Each strain was back crossed to wild type at least three times. A list of strains used in this study is provided in Table S4.

### CRISPR mutants
*set-2(syb1554) gfp::set-2* and *set-16(syb1762) gfp::set-16* strains were generated by SunyBiotech using CRISPR-Cas9 technology.

set-2(zr1504) carries a mutation in the SET domain (H1447K) described by Abay-Nørgaard et al. (2020). *set-16(zr1804)* carries a mutation in the SET domain (H2410K) while *set-16(zr1984)* recapitulates a point mutation in the *SETD1B (KMT2G)* gene, found in humans with intellectual disability (Hiraide et al., 2018). Briefly, ssDNA repair templates were designed with the desired mutation and three or four silent mutations to avoid re-cutting. The specific sgRNA sequences was cloned into the pJJR50 vector: *set-16(zr1804)*, TCTTGAGCAGTTCGGATCACACGA; *set-16(zr1984)*, TCTTGTCACCCATTCCTCGTCGATT.

A final mix containing the *pha-1* repair template and the vector pJW1285 (expressing the Cas9) was injected into *pha-1(e2123)* mutants at a concentration of 50 ng/µl for all templates. Wild-type animals were selected at 25°C and the presence of the mutation was confirmed by sequencing.

### RNAi
RNAi experiments were performed as described by Kamath et al. (2001). Wild-type L1 larvae were plated on NGM/IPTG plates and grown to the adult stage, then animals were treated with hypochlorite to collect the embryos used for protein extraction and western blot analyses. The *set-16(RNAi)* clone was obtained from a publicly available RNAi bacterial library (Kamath and Ahringer, 2003).

### Degron strains and auxin treatment
*set-2(syb4167)*, a strain with a degron-GFP cassette at the 5′ end of the endogenous *set-2* gene, was created by SunyBiotech. *set-2(syb4167)* was crossed into strains containing the *IeSi38* transgene, expressing modified *Arabidopsis thaliana TIR1* in the germline, and the *IeSi57* transgene, expressing modified *A. thaliana TIR1* in somatic tissues, to allow SET-2 degradation in all tissues after auxin treatment. Auxin treatment was performed as described by Zhang et al. (2015). Briefly, auxin indole-3-acetic acid solution was diluted in ethanol and added to NGM media at a final concentration of 1 mM before pouring the plates. Synchronised animals were kept at 20°C for 3 or 6 h in auxin plates to reach the young adult stage and collected by either picking or washed off the plates with M9 buffer.

### Western blot analyses
Proteins were extracted from embryos obtained by hypochlorite treatment. Samples were boiled for 5 min at 95°C in a solution of LDS sample buffer (1×) and 25 mM dithiothreitol. NuPAGE Bis-Tris Mini Protein Gels, 4-12%, 1.0-1.5 mm (Invitrogen, NP0335BOX) were used to separate proteins. Gels were run for 2 h at 120 V in freshly prepared NuPAGE MES SDS Running Buffer (20×- NP0002). Protein transfers were carried out using 1× Tris-Glycine buffer in a wet tank transfer systems at 100 V for 1 h. Nitrocellulose membranes were blocked in 5% non-fat milk at room temperature for 1 h. Primary antibodies used were: anti-H3K4me3 (Cell Signaling Technology, C42D8; 1:1000); anti-H3K4me2 (Invitrogen, 710796, 1:750), anti-H3K4me1 (Abcam, ab8895, 1:1000), polyclonal anti-H3 (Abcam, ab1791; 1:5000); and peroxidase-labelled anti-rabbit secondary antibody (Vector Laboratories, PI-1000-1; 1:10,000). Primary and secondary antibodies were diluted in blocking solution. Quantification was performed using ImageJ [National Institutes of Health (NIH)].

### Immunofluorescence
Embryo staining was performed as described by Chin-Sang et al. (1999). Briefly, after hypochlorite treatment, embryos were resuspended in a solution of 2× MRWB (Modified Ruvkun's Witches Brew) with 6% paraformaldehyde and immediately frozen in liquid nitrogen. The samples were then defrosted on ice and fixed in the same solution for at least 20 min. Primary antibodies were incubated overnight at 4°C; secondary antibodies were incubated for 2 h at room temperature.

Gonad staining was performed in extracted gonads from young adult animals. Extracted gonads were fixed for 10 min with 6% paraformaldehyde and immediately freeze-cracked. Slides were incubated with methanol for 5 min and washed three times with phosphate-buffered saline (PBS) and Tween-20 0.01% (PBST). Slides were blocked for 1 h at room temperature in 1% BSA (bovin serum albumin) in PBST. Primary antibodies were incubated overnight at 4°C in a humid chamber, and secondary antibodies were incubated for 2 h at room temperature. Each sample was stained with DAPI (Sigma-Aldrich, D9542) in a concentration of 0.1 ng/ml and the slides were mounted with Mowiol mounting media. The following primary antibodies were used: anti-H3K4me3 (Cell Signaling Technology, C42D8; 1:500); anti-H3K4me2 (Invitrogen, 710796; 1:500), anti-H3K4me1 (abcam, ab8895; 1:500), anti-PGL-1 (Developmental Studies Hybridoma Bank, K76; 1:10), anti-GFP (Life Technologies, A6455; 1:2000). The following secondary antibodies were used: goat anti-rabbit IgG Alexa Fluor

568 (Invitrogen, A11036; 1:500) and donkey anti-mouse IgG Alexa Fluor 488 (Invitrogen, A21202; 1:500). Primary and secondary antibodies were prepared in blocking solution. Slides were washed three times in PBST after each incubation. Images were taken using a Zeiss AXIO Imager M2 fluorescence microscope.

### Chromatin extraction and ChIP-seq

Young adult animals were selected for these analyses as they can be highly synchronised. Chromatin extraction from 10,000 animals per sample was performed as described by Zaghet et al. (2021). Briefly, animals (at young adult stage) were collected and fixed for 10 min with a solution of 1% formaldehyde. Sonication was performed using Diagenode bioruptor sonicator, with the following conditions: 12-15 cycles, 30 s on 30 s off, high sonication. Chromatin extraction from a small animal population (in auxin experiments) was performed following the protocol described at https://www.covaris.com/wp/wp-content/uploads/resources_pdf/M020069.pdf. One-thousand animals (at young adult stage) per sample were manually collected and fixed for 30 min with a solution of 2% formaldehyde. Sonication was performed using the Covaris E220evolution with the following conditions: PIP: 175, DF: 10%, CPB: 200 10-15 min. The following antibodies were used: anti-H3K4me3 (abcam, ab8580; 1:500) and anti H3K4me1 (abcam, ab8895; 1:500). Validation results of antibodies used in ChIP are available in an antibody validation database (http://compbio.med.harvard.edu/antibodies/). Samples were de-crosslinked overnight at 65°C. DNA was purified with PCR Mini Elute PCR purification kit (QIAGEN, 28006) and DNA concentration was measured using a Qubit assay. Libraries were prepared using NEBNext Ultra II DNA Library Prep Kit for Illumina (NEB, E7645S) and Multiplex Oligos for Illumina (NEB, E7335S, E7500S). The quality and size of the libraries were controlled using High Sensitivity DNA Kit (Agilent 2100 Bioanalyzer). Sequences were performed using a NextSeq 500 system and a NextSeq 500/550 High Output Kit v2 (Illumina, FC-404-2005).

### RNA extraction and RNA-seq

Animals grown at 20°C were collected at young adult stage and freeze-cracked in liquid nitrogen. RNA was extracted using Arcturus PicoPure RNA Isolation Kit (Thermo Fisher Scientific) and RNase-Free DNase Set (QIAGEN). Generally, four independent samples per genotype were prepared and analysed in parallel. RNA quality was tested using a Nano assay at Bioanalyzer (Agilent 2100 Bioanalyzer). Libraries were prepared using TruSeq RNA Library Prep Kit v.2 (Illumina, RS-122-2001) and size-checked using High Sensitivity DNA Kit (Agilent 2100 Bioanalyzer). Sequences were performed using a NextSeq 500 system and a NextSeq 500/550 High Output Kit v.2 (Illumina, FC-404-2005). Aligned reads were >40 M per sample.

### Bioinformatic analysis

ChIP-seq data were processed as described by Ho et al. (2014). Fastq files were trimmed for adaptor sequence and masked for low-complexity or low-quality sequence using trim_galore (version 0.6.4), then ChIP-seq trimmed fastq reads were aligned to *C. elegans* genome version WBcel235 using bowtie2 (v.2.3.5.1) with default parameters. Bam files were then sorted and indexed using samtools (v.1.10). ChIP enrichment was normalised by dividing to input using DeepTools bamCompare (v.3.3.0) with the following parameters: RPKM, bin-size of 10 bp, ignore duplicates, extend reads to 200 bp and remove ENCODE blacklisted regions (https://github.com/Boyle-Lab/Blacklist/). *C. elegans* enhancers were obtained from Jänes et al. (2018). Genome-wide data were visualised using the Integrative Genomics Viewer (v.2.6.1; Robinson et al., 2011).

For RNA-seq, the raw fastq files (stranded SE75) were trimmed for adaptor sequence and masked for low-complexity or low-quality sequence using trim_galore (v.0.6.4), then mapped to WBcel235 whole genome using HISAT 2.2.1 with default parameters. The number of reads mapped to genes was determined with featureCounts (v.1.6.4). FPKM stands for fragments per kilobase of transcript per million mapped reads at gene level. The resulting read count data were processed by DESeq2 (v.1.35.0) to identify DE transcripts among experimental groups. Genes with log2 fold change > ±1 and $P$adj <0.01 were considered to have significant expression change.

Plots of differential gene expression or H3K4me3 signals at TSS ±1 kb regions were visualised using ggplot2 and heatmap packages in R (v.4.0) with the indicated conditions.

GO analyses were performed using g-profiler (biit.cs.ut.ee/gprofiler/gost) with Bonferroni correction. Deregulated genes used for GO analysis had $P$<0.01 and log2FC±1.

### Quantitative PCR

cDNA was prepared using Maxima™ H minus cDNA synthesis master mix (Thermo Fisher Scientific). qPCR was performed using Lightcycler 480 SYBR Green I Master on a LightCycler480 (Roche). Gene expression was normalised to *pmp-3* and *cdc-42*. All reactions were performed in quadruplicate.

### Phenotypic analyses

Phenotypic analyses were conducted in animals kept in culture for five to seven generations.

### Tail defects

Three plates with ten young adult animals were kept at 20°C for 6 h, to allow the deposition of eggs. Adults were then removed and L1 larvae were collected in M9 the day after. Animals were immobilised using $NaN_3$ (0.25%), moved on a microscope slide with 5% agarose pad and analysed using a Zeiss AXIO Imager M2 fluorescence microscope. At least two biological independent experiments were performed.

### Gonad defects

Young adult animals were moved on a microscope slide, prepared with 5% agarose pad, and immobilised using $NaN_3$ (0.25%). Gonads were analysed using a Zeiss AXIO Imager M2 fluorescence microscope. Three biological independent experiments were performed.

### Brood size

Total progeny of at least ten animals per strain was counted. To perform the experiments at 20°C, single L4 larvae grown at 20°C were transferred to single plates, moved to new plates each day until no more eggs were laid and their progeny counted. To perform the experiments at 25°C, L4 larvae were moved from 20°C to 25°C and their progeny singled out at L4 stage, and their brood size was measured as described above.

### Mortal germline

L4 larvae were moved from 20°C to 25°C and the progeny produced was considered as P0. A total of eight lines per strain were analysed, with eight L4 larvae picked from the progeny of the previous generation and transferred to new plates. A line was considered sterile when fewer than ten animals were produced.

### Axon guidance defects

L4 animals were moved on a microscope slide, prepared with 5% agarose pad, and immobilised using $NaN_3$ (0.25%). The analysis was performed using a Zeiss AXIO Imager M2 fluorescence microscope. Three biological independent experiments were performed.

### Statistical analyses

Statistical analyses were performed using GraphPad Prism 9. The specific tests used to address statistical significance are stated in figure legends.

### Acknowledgements

We thank the *Caenorhabditis* Genetics Center, which is funded by the NIH Office of Research Infrastructure Programs (P40 OD010440); the National BioResource Project for *C. elegans* (Japan) and the International *C. elegans* Gene Knockout Consortium for providing strains. We thank the Biotech Research and Innovation Centre (BRIC) sequencing facility for its important contribution to this work.

### Competing interests

The authors declare no competing or financial interests.

### Author contributions

Conceptualization: B.A., K.H., S.A.-N., A.E.S.; Funding acquisition: S.A.-N., A.E.S.; Investigation: B.A., H.W., K.M., M.Z., S.A.-N., A.E.S.; Methodology: B.A., S.A.-N.,

A.E.S.; Supervision: K.H., A.E.S., S.A.-N.; Visualization: B.A., H.W., S.A.-N., A.E.S.; Writing – original draft: B.A., K.H., S.A.-N., A.E.S.; Writing – review & editing: B.A., H.W., K.M., M.Z., K.H., S.A.-N., A.E.S.

## Funding
This work was supported by the Danish National Research Foundation (Danmarks Grundforskningsfond; 118119 to A.E.S.), startup funds from the Peking University Health Science Center, the National Natural Science Foundation of China (to H.W.), a Memorial Sloan-Kettering Cancer Center support grant (NIH P3100 CA008748), a Tri-Institutional Stem Cell grant (Memorial Sloan Kettering Cancer Center, Rockefeller University and Weill-Cornell Medical College; 2019-035), and startup funds from The Institute of Cancer Research (UK) to K.H. Open Access funding provided by the Biotech Research and Innovation Centre, University of Copenhagen, Denmark. Deposited in PMC for immediate release.

## Data and resource availability
RNA-seq and ChIP-seq data have been deposited in Gene Expression Omnibus under accession number GSE282259. All other relevant data and details of resources can be found within the article and its supplementary information.

## Peer review history
The peer review history is available online at https://journals.biologists.com/dev/lookup/doi/10.1242/dev.204924.reviewer-comments.pdf

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
