## [Peer Review File · Development (Cambridge, England)]

Functional dissection of H3K4 methyltransferases reveals distinct catalytic and non-catalytic roles in *C. elegans* development

Benedetta Attianese, Hua Wang, Katrine Madsen, Mandoh Zeijdner, Kristian Helin, Steffen Abay-Noergaard and Anna Elisabetta Salcini

DOI: 10.1242/dev.204924

Editor: Haruhiko Koseki

Review timeline

Original submission:	7 May 2025
Editorial decision:	11 June 2025
First revision received:	17 July 2025
Editorial decision:	21 August 2025
Second revision received:	10 September 2025
Accepted:	27 September 2025

Original submission

First decision letter

MS ID#: dev.204924

MS TITLE: Functional dissection of H3K4 methyltransferases reveals distinct catalytic and non-catalytic roles in *C. elegans* development

AUTHORS: Benedetta Attianese, Hua Wang, Katrine Madsen, Mandoh Zeijdner, Kristian Helin, Steffen Abay-Noergaard and Anna Elisabetta Salcini

Dear Dr Abay-Noergaard,

I have now received all the referees' reports on the above manuscript, and have reached a decision. The referees' comments are appended below, or you can access them online: please go to:

As you will see, the referees express considerable interest in your work, but have some significant criticisms and recommend a substantial revision of your manuscript before we can consider publication. If you are able to revise the manuscript along the lines suggested, which may involve further experiments, I will be happy receive a revised version of the manuscript. Your revised paper will be re-reviewed by one or more of the original referees, and acceptance of your manuscript will depend on your addressing satisfactorily the reviewers' major concerns. Please also note that Development will normally permit only one round of major revision. If it would be helpful, you are welcome to contact us to discuss your revision in greater detail. Please send us a point-by-point response indicating your plans for addressing the referees' comments, and we will look over this and provide further guidance.

Please attend to all of the reviewers' comments and ensure that you clearly highlight all changes made in the revised manuscript. Please avoid using 'Tracked changes' in Word files as these are lost in PDF conversion. I should be grateful if you would also provide a point-by-point response detailing how you have dealt with the points raised by the reviewers in the 'Response to Reviewers' box. If you do not agree with any of their criticisms or suggestions please explain clearly why this is so.

Reviewer 1*Advance summary and potential significance to field*

Methylation of histone H3K4 is catalyzed by conserved HMTs of the SET1/MLL family. Attianese et al. investigate the catalytic and non-catalytic roles of *C. elegans* SET-2, the single homologue of SET1/COMPASS, and SET-16, most closely related to mammalian MLL1. They create catalytically inactive mutants for each by changing conserved residues in the catalytic domain, and show that both HMTs have functions independent of catalytic activity. The work is timely because it is becoming increasingly clear that histone methyltransferases play important functions independent of their enzymatic activity. However, claims of novelty need to be toned down throughout because a catalytically dead allele of SET-2 has already been published and characterized, as has the effect of SET-2 dependent H3K4me3 on genome-wide methylation patterns. The lack of correlation between set-2 dependent H3K4me3 and transcription of the underlying genes was also already reported.

The manuscript should therefore be completely reorganized, moving some main figures to supplementary, and others from supplementary to main to concentrate more on novel results regarding set-16, rather than those that confirm previous results. Additional clarifications are required.

Specific points are addressed below

Comments for the author

Specific comment

Abstract should be rewritten to emphasize novel results only. Specifically:

-"Gene expression profiling showed that lack of SET-2 and SET-16 catalytic activities results in deregulation of gene expression independent of H3K4me3 status at transcription start sites".

SET-2 is responsible for the majority of global H3K4me3, and the lack of correlation between set-2 dependent H3K4me3 and gene expression has already been shown for a set-2 null mutant (Beurton et al. 2019 DOI: 10.1093/nar/gkz880).

-"we... found that the catalytic activity of SET-2 is essential for proper development and fertility, whereas SET-16 enzymatic activity has cell-type-specific roles".

That SET-2 catalytic activity is essential for fertility was reported in Caron et al 2021 doi: 10.26508/lsa.202101140

-".. our results reveal that loss of the majority of H3K4me3 is compatible with development and reproduction in *C. elegans*"

See above comment. Also this is not fully accurate since as the authors themselves also show, loss of H3K4me3 is associated with reduced fertility

Results

-Line 111 : "Previous studies indicated that SET-2 is responsible for H3K4me2/3 deposition... The contribution of its catalytic activity to its biological functions remains, however, largely untested."

This is not correct. That fertility is dependent on SET-2 catalytic activity was already shown in Caron Life Sci Alliance 2021. They show that set-2(syb2085) animals carrying a catalytically dead allele show global loss of H3K4 di- and trimethylation (H3K4me2/3) (by WB analysis), reduced fertility, and sterility over subsequent generations

-line141 : effects of catalytic KO allele were comparable to the ones detected in set-2(tm1630), a bona fide null mutant

this has also been shown for the bn129 null mutant in Xiao Y. et al 2011 PNAS

-Figure 1: Both Li et al 2011 and Xiao et al 2011 showed differential contribution of SET-2 to H3K4me2 and me3 in embryos (somatic tissue) and adults (mostly germline). Did authors perform WB on young adults? Why were WB performed on embryos and ChIP-seq on yg adults?
Figure 1D : how do you explain that set-2 catalytic KO has greater effect on H3K4me1 and me2 than deletion allele tm 1630 ? Is this difference real ?

-Fig 2 SET-2 expression was previously shown to be ubiquitous using a SET-2 ::GFP reporter (Simonet Dev Biol 2002). SET-16 FISH shows ubiquitous mRNA expression (Engert et al. PLoS Genetics 2018).
So these findings are not surprising

Fig. 3 ChIP-seq analysis

The results on SET-2 are as expected from published work using ChIP-seq to look at H3K4me3 genome-wide distribution by ChIP-seq on set-2 null mutants (Beurton et al. 2019). What is new here is the set-16 and set-2;set-16 results. Screen shots for a couple of genes should be included in the figure.

The authors identify a cluster (Cluster four) that shows an unexpected increase of H3K4me3 level at TSS specifically in set-2(zr1504) mutant, that is restored to wild-type levels in set-257 2(zr1504) set-16(zr1804) double mutant...

The authors should be aware of ChIP-seq analysis carried out on set-2 null mutants by Beurton et al 2019 PNAS. They showed that the observed increase observed in specific genomic regions is actually an artefact on normalization. When a spike-in control was used, this increase disappeared.

-line 316 : loss of H3K4me3 mainly results in gene upregulation in *C. elegans*

This statement should be toned down, since Beurton et al found a similar number of up- and down-regulated genes in set-2 lof mutants at the young adult stage. Similarly in other species loss of H3K4me3 is associated with both up- and down-regulation of genes. The authors also find this tendency with the set-2 degron

-Figure 4 : lack of correlation between loss of H3K4me3 and gene expression :

That SET-2 is responsible for most global H3K4me3 was already shown (see above). Since, as the authors and others have shown, SET-16 contributes little to H3K4me3, the lack of correlation is not surprising and consistent with published results in *C. elegans* and other species from yeast to mouse.

The authors have H3K4me1 ChIP-seq data for SET-16. Why wasn't this data further analyzed ? Does it reveal anything about regulatory elements ? Transcriptions ?

Acute depletion experiments Figure 5

- AID acute degradation of the two HMT is novel. Unfortunately however SET-16 ::AID is non-functional, so the analysis is limited to SET-2. Having this data could add novelty to this work

- AID of SET-2 reduces H3K4me3 but not me2...this is in contrast to what is observed in mutant embryos in figure 1. This should be commented...is it simply a question of stage (emb vs yg adults ?)

- ChIP-seq of auxin treated animals identifies ~8,000 genes with significantly decreased H3K4me3 at their TSSs (log2FC<-1) at TSSs. This represents more than 30% of all genes !

Do the authors have any explanation as to why so many more genes would respond to acute H3K4me3 depletion than to loss of SET-2 (in a null of catalytically inactive mutant). IGV tracts should be shown... how was peak calling performed? Or is it possible that the set-2degron encodes a protein with gain-of-function activity, or that is not properly regulated?

Given this large amount of genes it is not surprising that they overlap with SET-2 targets identified in the set-2(zr1504) mutant

- Once a gain, a correlation between changes in H3K4me3 levels at TSSs and gene expression is not found. While this is most likely to reflect the reality-H3K4me3 does not correlate with expression of underlying genes - the large number of SET-2 dependent H3K4me3 peaks identified (more than 30% of all genes) makes this comparison questionable

-line 146 the authors reproduce the mrt phenotype observed with another catalytically dead set-2 allele (Caron et al eLife). Therefore it is not accurate to state that extensive depletion of H3K4 methylation is compatible with reproduction in *C. elegans*

The Helin lab recently showed that H3K4me3 regulates RNA polymerase II promoter-proximal pause-release. PolII pausing has been shown in *C. elegans*. Could a similar experiment be carried out in the set-2 mutant? This would really add novelty.

- phenotypic analysis

Li et al. 2011 showed that set-16(RNAi) caused >80% embryonic lethality, with survivors growing up to be Dumpy (Dpy phenotype) and sterile adults. Homozygous embryos from animals heterozygous for the set-16 (gk438) deletion allele (i.e., Maternal+/Zygotic-), develop into Dpy and sterile adults, indicating substantial maternal rescue.

The lethality of set-16 null mutations was also observed by Andersen and Horvitz 2007 Development 134.

By contrast animals carrying the set-16(zr1804) catalytic-inactive allele are viable and fertile at all temperatures tested, suggesting that the above phenotypes are independent of SET-16 catalytic activity.

Can the authors speculate on what these HMT independent functions may be? It would be interesting to know if identified protein interactions are maintained in this allele, for example. How is this related to HMT independent functions identified for other MLL family members?

Reviewer 2

Advance summary and potential significance to field

This study provides a detailed analysis of the enzymatic contributions of two lysine methyltransferases, SET-2 and SET-16, to histone H3K4 methylation and gene expression in *C. elegans*. The authors show that SET-2 is primarily responsible for H3K4me3 deposition, while SET-16 mediates H3K4me1 and also contributes to H3K4me3. Although H3K4me3 is typically associated with active transcription, they find that gene expression changes in their mutants do not correlate with H3K4me3 levels at transcriptional start sites, suggesting that H3K4me3 may not play a primary instructive role in transcriptional regulation in this context. The observation that acute SET-2 depletion causes a broader loss of H3K4me3 than seen in catalytic-dead mutants points to possible compensatory mechanisms that act over time. In addition, phenotypic differences between catalytic-dead and null alleles of set-16 suggest non-catalytic functions for this enzyme. While the findings are largely consistent with previous work in other systems, this study provides a well-executed and informative analysis in a metazoan model that contributes to ongoing efforts to refine our understanding of H3K4 methylation in gene regulation.

Comments for the author

The findings are interesting and potentially important, but the discussion would benefit from a more comprehensive treatment of prior work in other model systems—particularly studies in yeast and *Drosophila* showing that H3K4 methylation is dispensable for viability. This additional context would help clarify how the *C. elegans* results advance our understanding of the role of H3K4 methylation in transcription and development. In addition, a discussion of potential mechanisms that compensate for the loss of H3K4me3 would strengthen the manuscript. The authors suggest that altered transcription may hinge on changes in chromatin conformation following loss of SET-2 and H3K4me3. It would be helpful to elaborate on what specific mechanisms might be involved. Are there known compensatory pathways that have been described in other systems?

In summary, this is a well-executed study, but the manuscript requires revision for clarity, consistency, and contextualization of the findings within the framework of previous research.

General points:

1. There were a variety of typographical errors in the paper, and some points were not supported by appropriate citations. One example of both appears on lines 486-491: "...could be expected considering the relevance of epigenetics in gametes (gametes) in maintaining germ cell immortality, in supporting totipotency in the zygote, and the initiation of embryogenesis in absence of active transcription ([add citations])."
 2. Inconsistent usage of the term catalytically inactive (vs. catalytic-inactive)—I recommend standardizing to catalytically inactive throughout the text.
 3. When DE genes are compared in the figures, was direction of change considered? If not, this analysis could miss important differences.
 4. Finally, please italicize all gene (and allele) names per *C. elegans* conventions—for example, *set-2(tm1630)* and *set-2(zr1504)* on lines 275-276.
-

First revisionAuthor response to reviewers' comments

We thank the reviewers for their careful evaluation of our manuscript and for the constructive comments, which have greatly improved the clarity, rigor, and contextual framing of our work. We appreciate the recognition of the study's strengths, and we have addressed all points raised through revisions to the text, figures, and supplementary materials.

In particular, we have expanded the discussion of relevant literature, corrected inconsistencies in terminology and formatting, and adjusted the interpretation of specific results to ensure accuracy. Where appropriate, we have added new analyses or provided additional explanations to support our conclusions. All changes made in response to individual comments are detailed below in our point-by-point responses.

We are grateful for the thoughtful feedback and hope that the revised version of the manuscript meets the reviewers' expectations.

Reviewer 1

1.

Abstract should be rewritten to emphasize novel results only. Specifically:

"Gene expression profiling showed that lack of SET-2 and SET-16 catalytic activities results in deregulation of gene expression independent of H3K4me3 status at transcription start sites".

SET-2 is responsible for the majority of global H3K4me3, and the lack of correlation between set-2 dependent H3K4me3 and gene expression has already been shown for a set-2 null mutant (Beurton et al. 2019 DOI: 10.1093/nar/gkz880).

"we... found that the catalytic activity of SET-2 is essential for proper development and fertility, whereas SET-16 enzymatic activity has cell-type-specific roles".

That SET-2 catalytic activity is essential for fertility was reported in Caron et al 2021 doi: 10.26508/lsa.202101140

".. our results reveal that loss of the majority of H3K4me3 is compatible with development and reproduction in *C. elegans*"

See above comment. Also this is not fully accurate since as the authors themselves also show, loss of H3K4me3 is associated with reduced fertility

Response

We thank the reviewer for these comments. We have revised the final sentence of the abstract to clarify that loss of H3K4me3 is compatible with life under normal growth conditions, as stated in

the results section. We feel the remainder of the abstract reflects the novelty of our work, including the generation of two new catalytically inactive mutants, creation of a novel double mutant carrying inactivated KMT2 members with near-complete H3K4me3 loss, and the development of *set-2* inducible degradation strain. Furthermore, we tested the effect of these mutations in a variety of phenotypes, extending the analysis beyond fertility tests.

Results

2.

-Line 111 : "Previous studies indicated that SET-2 is responsible for H3K4me2/3 deposition... The contribution of its catalytic activity to its biological functions remains, however, largely untested."

This is not correct. That fertility is dependent on SET-2 catalytic activity was already shown in Caron Life Sci Alliance 2021. They show that *set-2(syb2085)* animals carrying a catalytically dead allele show global loss of H3K4 di- and trimethylation (H3K4me2/3) (by WB analysis), reduced fertility, and sterility over subsequent generations

Response

We thank the reviewer for pointing this out. We have corrected the statement and now also cite Caron et al. 2021

3.

-line141 : effects of catalytic KO allele were comparable to the ones detected in *set-2(tm1630)*, a bona fide null mutant this has also been shown for the *bn129* null mutant in Xiao Y. et al 2011 PNAS

Response

We thank the reviewer for the observation and modified the text accordingly.

4.

-Figure 1: Both Li et al 2011 and Xiao et al 2011 showed differential contribution of SET-2 to H3K4me2 and me3 in embryos (somatic tissue) and adults (mostly germline). Did authors perform WB on young adults? Why were WB performed on embryos and ChIP-seq on yg adults?

Response

We appreciate this question. Embryos were used for WB due to robust chromatin extraction and reproducibility. For ChIP-seq, young adults were chosen for more precise synchronization and because they include both somatic and germline chromatin. While this introduces some variation, both datasets consistently show dramatic H3K4me3 reduction in mutants.

Below, a western blot shows H3K4me3 levels detected in YA wild type and mutant, comparable to the ones observed in embryos.

Figure R1: Western blot at young adult stage in *set-2(tm1630)*, *set-2(zr1504)* and *set-16(zr1804)*

5.

Figure 1D : how do you explain that *set-2* catalytic KO has greater effect on H3K4me1 and me2 than deletion allele *tm1630* ? Is this difference real ?

Response

We thank the reviewer for this observation. The catalytic mutant (*zr1504*) shows slightly greater H3K4me1/2 reduction than the null (*tm1630*), but the differences are not statistically significant (me1 $p=0.0593$, me2 $p=0.1397$).

6.

-Fig 2 SET-2 expression was previously shown to be ubiquitous using a SET-2 ::GFP reporter (Simonet Dev Biol 2002). SET-16 FISH shows ubiquitous mRNA expression (Engert et al. PLoS Genetics 2018). So these findings are not surprising

Response

We agree and we now show this part of the study in supplementary information and cite previous studies

7.

Fig. 3 ChIP-seq analysis

The results on SET-2 are as expected from published work using ChIP-seq to look at H3K4me3 genome-wide distribution by ChIP-seq on *set-2* null mutants (Beurton et al. 2019). What is new here is the *set-16* and *set-2;set-16* results. Screen shots for a couple of genes should be included in the figure.

Response

For space reasons, we show the screenshots in **Figure S4a**.

8.

The authors identify a cluster (Cluster four) that shows an unexpected increase of H3K4me3 level at TSS specifically in *set-2(zr1504)* mutant, that is restored to wild-type levels in *set-257 2(zr1504)* *set-16(zr1804)* double mutant...

The authors should be aware of ChIP-seq analysis carried out on *set-2* null mutants by Beurton et al 2019 PNAS. They showed that the observed increase observed in specific genomic regions is actually an artefact on normalization. When a spike-in control was used, this increase disappeared.

Response

We thank the reviewer for this observation. We have removed comments to Cluster 4, to avoid overinterpreting potential artifacts.

9.

-line 316 : loss of H3K4me3 mainly results in gene upregulation in *C. elegans*

This statement should be toned down, since Beurton et al found a similar number of up- and down-regulated genes in *set-2* lof mutants at the young adult stage. Similarly in other species loss of H3K4me3 is associated with both up- and down-regulation of genes. The authors also find this tendency with the *set-2* degenon

Response

We thank the reviewer for this comment. We are now commenting this cautiously.
10.

-Figure 4 : lack of correlation between loss of H3K4me3 and gene expression :

That SET-2 is responsible for most global H3K4me3 was already shown (see above). Since, as the authors and others have shown, SET-16 contributes little to H3K4me3, the lack of correlation is not surprising and consistent with published results in *C. elegans* and other species from yeast to mouse.

Response

We appreciate the reviewer's point and agree that the limited correlation between H3K4me3 levels and gene expression has been noted in earlier studies, particularly for *set-2* null mutants. However, our study offers several advances.

First, we generated and analyzed catalytic point mutants of both *set-2* and *set-16*, allowing us to precisely address the effect of the catalytic activity of KMT2 proteins on transcription, for the first time. Second, we established a double mutant (*set-2(zr1504); set-16(zr1804)*), enabling us to assess gene expression and chromatin structure in the context of near-total depletion of global H3K4me3 – a condition not previously examined in *C. elegans*. This genetic background provides a better context to test the relationship between H3K4me3 and transcription.

Finally, we employed an auxin-inducible degron system to achieve acute SET-2 depletion, offering temporal resolution that distinguishes immediate chromatin and transcriptional responses from longer-term compensatory adaptations. Together, these complementary tools provide a uniquely powerful platform to dissect both catalytic and non-catalytic functions of COMPASS components, and to clarify how transcription performs in the absence of canonical H3K4me3 marking.

11.

The authors have H3K4me1 ChIP-seq data for SET-16. Why wasn't this data further analyzed ? Does it reveal anything about regulatory elements ? Transcriptions ?

Response

We thank the reviewer for this suggestion. We performed H3K4me1 ChIP-seq in *set-16* mutants to validate its catalytic specificity and we chose to focus our analyses on H3K4me3 due to its broader enrichment at promoters and clearer associations with gene expression. The H3K4me1 data (presented in Supplementary Figure S3) do show reduced signal at enhancers and TSSs in *set-16(zr1804)*, consistent with a role of SET-16 as an H3K4me1 methyltransferase. A dedicated analysis of enhancer regulation by SET-16 would be valuable but beyond the scope of the current study.

12.

Acute depletion experiments Figure 5

- AID acute degradation of the two HMT is novel. Unfortunately however SET-16 ::AID is non-functional, so the analysis is limited to SET-2. Having this data could add novelty to this work
- AID of SET-2 reduces H3K4me3 but not me2...this is in contrast to what is observed in mutant embryos in figure 1. This should be commented...is it simply a question of stage (emb vs yg adults ?)

Response 12

We thank the reviewer for the comment. We agree that the inability to generate a viable SET-16::AID strain is a limitation that at the moment we cannot overcome. We state clearly in the main text and figure legends that Western blot and ChIP-seq were performed at different developmental stages (embryos vs. young

adults), which likely accounts for this difference.

13.

- CHIP-seq of auxin treated animals identifies ~8,000 genes with significantly decreased H3K4me3 at their TSSs ($\log_2FC < -1$) at TSSs. This represents more than 30% of all genes !

Do the authors have any explanation as to why so many more genes would respond to acute H3K4me3 depletion than to loss of SET-2 (in a null of catalytically inactive mutant). IGV tracks should be shown... how was peak calling performed? Or is it possible that the set-2degron encodes a protein with gain-of-function activity, or that is not properly regulated?

Given this large amount of genes it is not surprising that they overlap with SET-2 targets identified in the set-2(zr1504) mutant

- Once a gain, a correlation between changes in H3K4me3 levels at TSSs and gene expression is not found. While this is most likely to reflect the reality-H3K4me3 does not correlate with expression of underlying genes - the large number of SET-2 dependent H3K4me3 peaks identified (more than 30% of all genes) makes this comparison questionable

Response

As described in figure 5 legend, we do not use peak calling, but we focus on CeTSSs. We believe that the high peak number reflects the immediacy of depletion, capturing *set-2* dependent H3K4me3 loss before compensatory mechanisms can act. In the text (line 297) we have mistakenly written genes instead of peaks, now corrected.

While we cannot formally exclude that the degron SET-2 results in a protein not properly regulated, we found that

1) SET-2 is degraded upon auxin treatment (Fig 5A,B), 2) animals carrying degron-tagged SET-2 behave like wild-type, developing normally without auxin at 25C (see figure R3 below).

In figure 4R, we show representative IGV browser tracks. if considered important, we can include these tracks in a supplementary figure.

Figure R3: mrt assay as described in material and methods on SET-2:AID strain with auxin and without auxin (EtOH)

Figure R4: Representative IGV browser tracks from control(etoh) and auxin treated samples - both input and IP

14.

-line 146 the authors reproduce the *mrt* phenotype observed with another catalytically dead *set-2* allele (Caron et al eLife). Therefore it is not accurate to state that extensive depletion of H3K4 methylation is compatible with reproduction in *C. elegans*.

Response

We thank the reviewer for pointing this out. We performed the *mrt* assay at 25 °C, as established by Ahmed S et al. Nature 2000 recognised as a stressed condition for the animals. Indeed, we have stated in the text that extensive depletion of H3K4 methylation is compatible with reproduction in *C. elegans* under standard conditions (20 °C). We do not find a publication by Caron et al in Elife, and we assume that the publication cited is Caron et al, 2022 in LSA.

15.

The Helin lab recently showed that H3K4me3 regulates RNA polymerase II promoter-proximal pause-release. PolII pausing has been shown in *C. elegans*. Could a similar experiment be carried out in the *set-2* mutant? This would really add novelty.

Response

We appreciate the reviewer's insightful suggestion regarding RNA polymerase II pause-release regulation by H3K4me3 and, we agree that examining Pol II dynamics in *set-2* catalytic and degenon mutants would be a highly valuable future direction to further address the impact of H3K4 methylation of transcription in nematode.

16.

- phenotypic analysis

Li et al. 2011 showed that *set-16*(RNAi) caused >80% embryonic lethality, with survivors growing up to be Dumpy (Dpy phenotype) and sterile adults. Homozygous embryos from animals heterozygous for the *set-16* (*gk438*) deletion allele (i.e., Maternal+/Zygotic-), develop into Dpy and sterile adults, indicating substantial maternal rescue.

The lethality of *set-16* null mutations was also observed by Andersen and Horvitz 2007 Development 134.

By contrast animals carrying the *set-16*(*zr1804*) catalytic-inactive allele are viable and fertile at all temperatures tested, suggesting that the above phenotypes are independent of SET-16 catalytic

activity.

Can the authors speculate on what these HMT independent functions may be? It would be interesting to know if identified protein interactions are maintained in this allele, for example. How is this related to HMT independent functions identified for other MLL family members?

Response 16

We thank the reviewer for this comment. We extended our discussion on this interesting point in the revised version of the manuscript.

Reviewer 2

The findings are interesting and potentially important, but the discussion would benefit from a more comprehensive treatment of prior work in other model systems—particularly studies in yeast and *Drosophila* showing that H3K4 methylation is dispensable for viability. This additional context would help clarify how the *C. elegans* results advance our understanding of the role of H3K4 methylation in transcription and development. In addition, a discussion of potential mechanisms that compensate for the loss of H3K4me3 would strengthen the manuscript. The authors suggest that altered transcription may hinge on changes in chromatin conformation following loss of SET-2 and H3K4me3. It would be helpful to elaborate on what specific mechanisms might be involved. Are there known compensatory pathways that have been described in other systems?

In summary, this is a well-executed study, but the manuscript requires revision for clarity, consistency, and contextualization of the findings within the framework of previous research.

Response

We thank the reviewer for the positive comments regarding our study. The revised discussion includes the contextualization of our findings within the broader landscape of prior work and explores potential compensatory mechanisms that may sustain gene expression in the absence of H3K4me3. Changes in the text are underlined

General points:

1.

There were a variety of typographical errors in the paper, and some points were not supported by appropriate citations. One example of both appears on lines 486-491: "...could be expected considering the relevance of epigenetics in gametes (gametes) in maintaining germ cell immortality, in supporting totipotency in the zygote, and the initiation of embryogenesis in absence of active transcription ([add citations])."

Response

We thank the reviewer for pointing this out. We have added the citation at the correct place and corrected the spelling mistake

2.

Inconsistent usage of the term catalytically inactive (vs. catalytic-inactive)—I recommend standardizing to catalytically inactive throughout the text.

Response

We thank the reviewer for pointing this out. We have changed it to catalytically inactive throughout the manuscript

3.

When DE genes are compared in the figures, was direction of change considered? If not, this

analysis could miss important differences.

Response

We thank the reviewer for the thoughtful suggestion. In response, we performed a more detailed analysis comparing the directionality of gene expression changes across genotypes, shown below in Figure R5. This analysis revealed that more than 75% of differentially expressed genes in *set-2(zr1504)* change in the same direction as in the *set-2(tm1630)* null mutant, and that nearly 50% overlap directionally between *set-2* and *set-16* mutants.

Given the high degree of concordance, we believe this analysis largely reinforces the existing conclusions and does not add substantial new insights. However, we are happy to include the data as a supplementary figure if the editor or reviewer feels it would be beneficial

Figure R5: Venn diagram of UP and down genes in *zr1504* and *tm1630*, and between *set-2* and *set-16*

4.

Finally, please italicize all gene (and allele) names per *C. elegans* conventions—for example, *set-2(tm1630)* and *set-2(zr1504)* on lines 275-276.

Response 4

We thank the reviewer for pointing this out. We have changed the missing italics in the gene names and alleles.

Second decision letter

MS ID#: dev.204924R1

MS TITLE: Functional dissection of H3K4 methyltransferases reveals distinct catalytic and non-catalytic roles in *C. elegans* development

AUTHORS: Benedetta Attianese, Hua Wang, Katrine Madsen, Mandoh Zeijdner, Kristian Helin, Steffen Abay-Noergaard and Anna Elisabetta Salcini

Dear Dr Abay-Noergaard,

I have now received all the referees' reports on the above manuscript, and have reached a decision. The referees' comments are appended below, or you can access them online: please go to: *****

As you will see, the referees express considerable interest in your work, but still have some significant criticisms and recommend further revision of your manuscript before we can consider publication. If you are able to revise the manuscript along the lines suggested, which may involve further experiments, I will be happy to receive a revised version of the manuscript. Your revised paper will be re-reviewed by one or more of the original referees, and acceptance of your manuscript will depend on your addressing satisfactorily the reviewers' major concerns. Please also note that Development will normally permit only one round of major revision. If it would be helpful, you are welcome to contact us to discuss your revision in greater detail. Please send us a point-by-point response indicating your plans for addressing the referees' comments, and we will look over this and provide further guidance.

Please attend to all of the reviewers' comments and ensure that you clearly highlight all changes made in the revised manuscript. Please avoid using 'Tracked changes' in Word files as these are lost in PDF conversion. I should be grateful if you would also provide a point-by-point response detailing how you have dealt with the points raised by the reviewers in the 'Response to Reviewers' box. If you do not agree with any of their criticisms or suggestions please explain clearly why this is so.

Reviewer 1

Advance summary and potential significance to field

The authors have taken into account many of my previous suggestions, improving the MS. However, claims to the novelty of these results are still overstated. In particular

1) As previously mentioned, the Abstract should be rewritten to emphasize novel results only. This was not done. Specifically:

i) Gene expression profiling showed that lack of SET-2 and SET-16 catalytic activities results in deregulation of gene expression independent of H3K4me3 status at transcription start sites. Should be replaced with:

-Gene expression profiling showed that simultaneous inactivation of SET-2 and SET-16 catalytic activities results in gene deregulation independent of H3K4me3 status at transcription start sites

As mentioned in my previous review, lack of correlation between set-2 dependent H3K4me3 at promoters and gene expression has already been shown for a set-2 null mutant (Beurton et al. 2019 DOI: 10.1093/nar/gkz880). The novelty here is in the use of the double catalytic KO.

ii) Finally, we examined the relevance of SET-2 and SET-16 catalytic activity on phenotypes identified in null mutants and found SET-2 catalytic activity essential for proper development and fertility, whereas SET-16 enzymatic activity has cell-type-specific roles.

Should be modified:

-Finally, we examined the relevance of SET-2 and SET-16 catalytic activity on phenotypes identified in null mutants and found that SET-2 catalytic activity is essential for proper somatic development. SET-16 enzymatic activity instead has cell-type-specific roles.

As already mentioned, that SET-2 catalytic activity is essential for fertility was previously reported in Caron et al 2021 doi: 10.26508/lisa.202101140. So they confirm this finding, and show that catalytic activity is also required for the neuronal phenotypes they previously described.

2) Because it is already known that set-2 is responsible for most global H3K4me3, with SET-16 contributing very little if at all to this mark (the authors confirm this in Fig1), the following statement is not accurate:

Our results reveal catalytic-dependent and -independent roles of KMT2 members and that loss of most H3K4me3 is compatible with life in *C. elegans*.

Suggestion :

Our results reveal catalytic-dependent and -independent roles of KMT2 members, and that combined loss of SET-16 and SET-2 is compatible with life in *C. elegans*.

This would make the abstract more accurate and remove unsubstantiated claims of novelty

Other points which were not addressed from previous version need to be modified accordingly :

3) -Line 102: Previous studies indicated that SET-2 is responsible for H3K4me2/3 deposition, transcriptional regulation, and normal development, (Abay-NÅ, rgaard et al., 2020;104 Beurton et al., 2019; Caron et al., 2022; Greer et al., 2010; Han et al., 2017; Herbette105 et al., 2017; Robert et al., 2014), and its catalytic activity is required fertility (Caron LSA 2021). However, the impact of SET-2 catalytic activity on the genome-wide distribution of H3K4me3 and transcription has not been systematically investigated .

As already previously noted, that fertility is dependent on SET-2 catalytic activity was already shown in Caron Life Sci Alliance 2021.

4) Expression patterns of SET-2 and SET-16

The phrase

Our findings confirm and extend previous reports of broad SET-2 and SET-16 expression (Engert et al., 2018; Simonet et al.,2007)

should be at the beginning of the section rather than the end, as there is very little that is new in this section

5) Fig. 3 Representative IGV browser tracks are shown in the authors response (RV). Is this set-2 acute depletion? This figure should be included in supplementary, also including a region where H3K4me3 is NOT affected and screen shots of the the set-2, set-16 double

6) -Figure 4 : lack of correlation between loss of H3K4me3 and gene expression :

My previous comment was that we already know that SET-2 is responsible for most global H3K4me3, with SET-16 contributing little if any. The authors themselves confirm this in figure 1

In their response, they claim that their study study offers several advances.

First, we generated and analyzed catalytic point mutants of both set-2 and set-16, allowing us to precisely address the effect of the catalytic activity of KMT2 proteins on transcription, for the first time. Second, we established a double mutant (set-2(zr1504); set-16(zr1804)), enabling us to assess gene expression and chromatin structure in the context of near-total depletion of global H3K4me3 – a condition not previously examined in *C. elegans*. This genetic background provides a better context to test the relationship between H3K4me3 and transcription..

As shown in Figure 1 and 4, set-2 is the main contributor to H3K4me3 by large, and its not clear to me to what extent set-16 contributes to further reducing H3K4me3 at SET-2 targets

7) Line 204 H3K4me3 signal was nearly abolished in the set-2(zr1504) set-16(zr1804) double mutant, at TSSs and putative enhancers (Fig.3A), indicating that SET-2 and SET-16

jointly catalyse the majority of H3K4me3 at transcriptional regulatory regions.

What the figure shows is that SET-2 catalyzes the majority of H3K4me3 and SET-16 has a minor contribution

Reviewer 2

Advance summary and potential significance to field

This paper addresses the role of enzymatic activity in two KMT methyltransferases in *C. elegans* - SET-2 and SET-16. The advances include the generation of a catalytically inactive SET-16 and a double mutant lacking both SET-2 and SET-16 catalytic activities, resulting in the loss of all KMT methyltransferase activity. In addition, acute depletion of SET-2 enables identification of the immediate targets of the SET-2 KMT, in contrast to long-term inactivation.

Comments for the author

By and large, the authors have addressed the previous critiques. I remain concerned about the significant overlap between the data presented here and what has already been published by other groups using similar alleles. This concern would be lessened by the inclusion of SET-16 AID inactivation. The authors state they were unable to generate a degron-GFP-tagged SET-16 using the traditional AID system. Were any attempts made to generate this strain using the improved AID2 system (Negishi, 2022)? This approach reduces auxin-independent gene inactivation and lowers the amount of auxin required to treat animals.

A related question: was an experiment performed to assess the effects of auxin itself (independent of SET-2 degradation) on GE and methylation states?

I have indicated specific editing suggestions below:

Figure 1C-D: It would be useful to have bar graphs showing each methylation state in all mutants with statistical comparison (e.g. H3K4me3 in *set-2(tm1630)* vs. *set-2(zr1504)* vs. double). This could be a supplemental figure.

Figure S2: embryonic precursor germ cells precursor (usually primordial germ cells).

Line 132-33: comparable to the ones detected in *set-2(tm1630)* and another null mutant alleles previously described *set-2(bn129)* (cites).

Line 162: substantial - should be substantial

Line 176: embryonic precursor germ cells. Typical nomenclature is primordial germ cells.

Line 238: share a consistent number of DE genes (489 genes). Do you mean considerable?

Lines 293-4: remained largely unchanged (Fig. 5B) was observed.

Line 328, 401, 414, 418: Italicize gene/allele.

Line 453-455: It is possible that their altered transcription hinges on changes of tin conformation occurring after loss of SET-2 and H3K4me3. I don't understand this sentence.

Second revision

Author response to reviewers' comments

Response to reviewers

We sincerely thank the reviewers for their careful re-evaluation of our revised manuscript and for their constructive feedback. We are pleased that the reviewers found our revisions have improved the clarity and rigor of the work. In this second round of review, we have carefully addressed all remaining points through targeted textual changes, clarification of figures, and corrections of errors. Where feasible, we have expanded our analyses or added supplemental information to strengthen the manuscript. We believe the revised manuscript now fully addresses the essential

concerns and provides a clear and accurate account of the novelty and impact of our findings.

Underneath is a detailed point-by-point response to reviewers comments

Reviewer 1

1) As previously mentioned, the Abstract should be rewritten to emphasize novel results only. This was not done. Specifically:

i) Gene expression profiling showed that lack of SET-2 and SET-16 catalytic activities results in deregulation of gene expression independent of H3K4me3 status at transcription start sites.

Should be replaced with:

-Gene expression profiling showed that simultaneous inactivation of SET-2 and SET- 16 catalytic activities results in gene deregulation independent of H3K4me3 status at transcription start sites

As mentioned in my previous review, lack of correlation between set-2 dependent H3K4me3 at promoters and gene expression has already been shown for a set-2 null mutant (Beurton et al. 2019 DOI: 10.1093/nar/gkz880). The novelty here is in the use of the double catalytic KO.

ii) Finally, we examined the relevance of SET-2 and SET-16 catalytic activity on phenotypes identified in null mutants and found SET-2 catalytic activity essential for proper development and fertility, whereas SET-16 enzymatic activity has cell-type- specific roles.

Should be modified:

-Finally, we examined the relevance of SET-2 and SET-16 catalytic activity on phenotypes identified in null mutants and found that SET-2 catalytic activity is essential for proper somatic development. SET-16 enzymatic activity instead has cell-type-specific roles.

As already mentioned, that SET-2 catalytic activity is essential for fertility was previously reported in Caron et al 2021 doi: 10.26508/lsa.202101140. So they confirm this finding, and show that catalytic activity is also required for the neuronal phenotypes they previously described.

2) Because it is already known that set-2 is responsible for most global H3K4me3, with SET-16 contributing very little if at all to this mark (the authors confirm this in Fig1), the following statement is not accurate:

Our results reveal catalytic-dependent and -independent roles of KMT2 members and that loss of most H3K4me3 is compatible with life in *C. elegans*.

Suggestion :

Our results reveal catalytic-dependent and -independent roles of KMT2 members, and that combined loss of SET-16 and SET-2 is compatible with life in *C. elegans*.

This would make the abstract more accurate and remove unsubstantiated claims of novelty

Response for abstract

We thank the reviewer for these constructive comments and we have changed the abstract accordingly.

Other points which were not addressed from previous version need to be modified accordingly :

3) -Line 102: Previous studies indicated that SET-2 is responsible for H3K4me2/3 deposition, transcriptional regulation, and normal development, (Abay-Nørgaard et al., 2020;104 Beurton et al., 2019; Caron et al., 2022; Greer et al., 2010; Han et al., 2017; Herbet105 et al., 2017; Robert et al., 2014), and its catalytic activity is required fertility (Caron LSA 2021). However, the impact of SET-2 catalytic activity on the genome-wide distribution of H3K4me3 and transcription has not been systematically investigated .

As already previously noted, that fertility is dependent on SET-2 catalytic activity was already shown in Caron Life Sci Alliance 2021.

Response

We thank the reviewer for this comment and we have changed the manuscript accordingly.

4) Expression patterns of SET-2 and SET-16

The phrase

Our findings confirm and extend previous reports of broad SET-2 and SET-16 expression (Engert et al., 2018; Simonet et al., 2007) should be at the beginning of the section rather than the end, as there is very little that is new in this section

Response

We have now moved the statement to the beginning of the section as requested. We however like to specify that in this section we show the expression of SET-16 in germline that, which we believe has not be reported before.

5) Fig. 3 Representative IGV browser tracks are shown in the authors response (RV). Is this set-2 acute depletion? This figure should be included in supplementary, also including a region where H3K4me3 is NOT affected and screen shots of the the set-2, set-16 double

Response

The tracks presented in RV are from the analysis of SET-2 acute depletion. We have now included them in Supplementary Figure 4C, alongside with a not affected region. IGV browser screenshots from the double mutant is reported in Figure S4A.

6) -Figure 4 : lack of correlation between loss of H3K4me3 and gene expression :

My previous comment was that we already know that SET-2 is responsible for most global H3K4me3, with SET-16 contributing little if any. The authors themselves confirm this in figure 1

In their response, they claim that their study offers several advances.

First, we generated and analyzed catalytic point mutants of both set-2 and set-16, allowing us to precisely address the effect of the catalytic activity of KMT2 proteins on transcription, for the first time. Second, we established a double mutant (set-2(zr1504); set- 16(zr1804)), enabling us to assess gene expression and chromatin structure in the context of near-total depletion of global H3K4me3 – a condition not previously examined in *C. elegans*. This genetic background provides a better context to test the relationship between H3K4me3 and transcription..

As shown in Figure 1 and 4, set-2 is the main contributor to H3K4me3 by large, and its not clear to me to what extent set-16 contributes to further reducing H3K4me3 at SET-2 targets

Response

We thank the reviewer for this important point. We agree that SET-2 is the major contributor to global H3K4me3, as shown in both our western blot (Figure 1) and ChIP-seq analyses (Figure 3). The contribution of SET-16 to overall H3K4me3 levels is indeed modest by comparison as shown in Figure 3B,D,E and we now state that in the text L208-209

7) Line 204 H3K4me3 signal was nearly abolished in the set-2(zr1504) set- 16(zr1804) double mutant, at TSSs and putative enhancers (Fig.3A), indicating that SET-2 and SET-16 jointly catalyse the majority of H3K4me3 at transcriptional regulatory regions.

What the figure shows is that SET-2 catalyzes the majority of H3K4me3 and SET-16 has a minor contribution

Response

We agree that our data show that SET-2 is the predominant H3K4me3 methyltransferase, with SET-16 contributing only modestly by comparison. We have revised the sentence on line 204 to better reflect this, as follows: These data confirms that SET-2 catalyzes the majority of H3K4me3 and SET-16 has a minor contribution (now line 208-209)

Reviewer 2

By and large, the authors have addressed the previous critiques. I remain concerned about the significant overlap between the data presented here and what has already been published by other groups using similar alleles. This concern would be lessened by the inclusion of SET-16 AID inactivation. The authors state they were unable to generate a degraon-GFP-tagged SET-16 using the traditional AID system. Were any attempts made to generate this strain using the improved AID2 system (Negishi, 2022)? This approach reduces auxin-independent gene inactivation and lowers the amount of auxin required to treat animals.

Response

We thank the reviewer for raising this important point. We agree that acute inactivation of SET-16 would provide valuable complementary insight. As noted, our attempts to generate a degraon-GFP-tagged SET-16 allele using the traditional AID system were unsuccessful, likely reflecting the essential roles of SET-16 during early development. We have not yet attempted to generate an allele using the recently described AID2 system (Negishi et al., 2022), and this might represent a promising future strategy to investigate SET-16 activity at different stages of development and in adult animals. It should be noted, however, that a substantial gene-specific basal degradation's been recently observed also in the AID2 system (De Xing, Nature Com 2025). Despite a new system AID2.1 has been suggested to improve this problem, this novel tool has not been tested in C. elegans. This approach will require the generation of new strains (hopefully successful) and ChIP and RNA sequencing datasets, which cannot be accomplished within the timeframe of a second revision.

A related question: was an experiment performed to assess the effects of auxin itself (independent of SET-2 degradation) on GE and methylation states?

Response

We thank the reviewer for raising this interesting point. We did not evaluate if auxin itself has any effect on transcription or chromatin status based on the fact that a) RNA seq analysis in tir-1 background (both germline and soma) revealed that very few genes (<35) show differential expression after auxin exposure (Morphis et al 2022) and b) Low doses of auxin for a limited time (1mM for few hours, as used in our approach) are reported not to be influential on brood size, embryonic viability, developmental rate and chromosomal segregation (e.g. Zhang et al 2015). These results suggest that transcription and chromatin organisation are not strongly influenced by auxin, at least at the condition used in our experiments. However, we believe that the effect of auxin on transcription and chromatin organization should be evaluated when performing long exposure to auxin (days), as it has been shown to extend lifespan and protect animals from endoplasmic reticulum stress (Loose and Ghasi, 2021, Bhoi et al. 2021).

I have indicated specific editing suggestions below:

Figure 1C-D: It would be useful to have bar graphs showing each methylation state in all mutants with statistical comparison (e.g. H3K4me3 in set-2(tm1630) vs. set- 2(zr1504) vs. double). This could be a supplemental figure.

Response

We agree with the reviewer and have now added a bar graph with statistical comparison between mutants at each methylation stage in figure S1A

Figure S2: embryonic precursor germ cells precursor (usually primordial germ cells).

Response

We agree with the reviewer and have now changed it in the text.

Line 132-33: comparable to the ones detected in set-2(tm1630) and another null mutant alleles previously described set-2(bn129) (cites).

Response

We agree with the reviewer and have now changed it in the text.

Line 162: substancial - should be substantial

Response

We agree with the reviewer and have now changed it in the text.

Line 176: embryonic precursor germ cells. Typical nomenclature is primordial germ cells.

Response

We agree with the reviewer and have now changed it in the text.

Line 238: share a consistent number of DE genes (489 genes). Do you mean considerable?

Response

We thank the reviewer for the correction - considerable is the right word and has been changed in the text.

Lines 293-4: remained largely unchanged (Fig. 5B) was observed.

Response

We thank the reviewer for the correction, was observed has been removed from the text.

Line 328, 401, 414, 418: Italicize gene/allele.

Response

We thank the reviewer for the correction, and have now italicized the gene/allele names.

Line 453-455: It is possible that their altered transcription hinges on changes of tin conformation occurring after loss of SET-2 and H3K4me3. I don't understand this sentence.

Response

We thank the reviewer for observing this mistake, and the text was changed to:

It is possible that altered transcription hinges on changes of chromatin conformation occurring after loss of SET-2 and H3K4me3. This is now been changed in the text.

Third decision letter

MS ID#: dev.204924R2

MS TITLE: Functional dissection of H3K4 methyltransferases reveals distinct catalytic and non-catalytic roles in *C. elegans* development

AUTHORS: Benedetta Attianese, Hua Wang, Katrine Madsen, Mandoh Zeijdner, Kristian Helin, Steffen Abay-Noergaard and Anna Elisabetta Salcini

Dear Dr Abay-Noergaard,

I am happy to tell you that your manuscript has been accepted for publication in Development, pending our standard publication integrity checks.

Reviewer 2

Advance summary and potential significance to field

The authors have fully addressed the concerns raised in the last round of reviews. The revised abstract incorporates the suggested changes and no longer overstates the novelty of the developmental findings or the conclusions regarding H3K4me3. I am satisfied that the previous criticisms have been addressed.